# Implementation of a new crop and irrigation scheme in the ISBA land surface model using SURFEX_v8.1

Arsène Druel[1,2], Simon Munier[1], Anthony Mucia[1], Clément Albergel[1,3] and Jean-Christophe Calvet[1]

[1]CNRM, Université de Toulouse, Météo-France, CNRS, Toulouse, France

[2]Now at Ecologie des Forêts Méditerranéennes (URFM), Institut national de recherche pour l'agriculture, l'alimentation et l'environnement (INRAE), Avignon, France

[3]Now at European Space Agency Climate Office, ECSAT, Harwell Campus, OX11 0FD Didcot, Oxfordshire, United Kingdom

*Correspondence to*: Jean-Christophe Calvet (jean-christophe.calvet@meteo.fr), Arsène Druel
(arsene.druel@umr-cnrm.fr)

**Abstract.** With an increase in the number of natural processes represented, global land surface models (LSMs) have become more and more accurate in representing natural terrestrial ecosystems. However, they are still limited with respect to the impact of agriculture on land surface variables. This is particularly true for agro-hydrological processes related to a strong human control on freshwater. While
most LSMs consider natural processes only, the development of human-related processes, e.g. crop phenology and irrigation in LSMs, is key. In this study we present the implementation of a new crop and irrigation scheme in the ISBA (Interaction between Soil, Biosphere, and Atmosphere) LSM. This highly flexible scheme is designed to account for various configurations and can be applied at different spatial scales. For each vegetation type within a model grid cell, three irrigation systems can be used at
the same time. A limited number of parameters are used to control (1) the amount of water used for irrigation, (2) irrigation triggering (based on the soil moisture stress) and (3) crop seasonality (emergence, harvesting). A case study is presented over Nebraska (USA). This region is chosen for its high irrigation density and because independent observations of irrigation practices can be used to verify the simulated irrigation amounts. The ISBA simulations with and without the new crop
phenology and irrigation scheme are compared to different satellite-based observations. The comparison shows that the irrigation scheme improves the simulated vegetation variables such as leaf area index and gross primary productivity and other variables largely impacted by irrigation such as evapotranspiration and land surface temperature. In addition to a better representation of land surface processes, the results

point to potential applications of this new version of the ISBA model for water resource monitoring and climate change impact studies.

## 1 Introduction

Amongst the global water withdrawal from rivers, reservoirs and groundwater, the share used for agriculture is estimated to reach 69 % on average, with some regional heterogeneity - over 90 % in some regions (Hoekstra and Mekonnen, 2012, FAO, 2014). This amount of water is likely to increase in the future in relation to climate warming and population growth (United Nations et al., 2019, Field et al., 2014). Future irrigation needs will likely be stronger in Africa. Now, only 5 % of cultivated land is under irrigation in Africa, against 21 % at a global scale (FAO, 2014). The historical evolution of irrigation also points to increasing water consumption: the area equipped for irrigation nearly doubled from 1900 to 1950, when it tripled from 1950 to 2005 (Siebert et al., 2015).

Irrigation is used to increase crop yields by controlling the soil water stress (Fraiture et al., 2007). Several studies indicate that yields can be higher by a factor of two or more when the fields are irrigated (Bruinsma, 2009; Colaizzi et al., 2009; Siebert and Döll, 2010; FAO, 2014). However, freshwater is already a limited resource and the current evolution of irrigation has a substantial impact on: (1) river discharge, with a decrease in their lower reaches due to diversions and impoundments for irrigation (Tang et al., 2008; Piao et al., 2010; Grafton et al., 2018), (2) groundwater level, with critical low levels observed in case of intensive irrigation (Rodell et al., 2009; Döll et al., 2012; Pfeiffer and Lin, 2014), (3) the surface energy budget through an increase of evapotranspiration, which can lead to surface cooling (Kueppers et al., 2007; Lobell et al., 2008; Jiang et al., 2014; de Vrese et al., 2016). Water vapour originating from large scale irrigation water supply can be recycled to rainfall and affect non-irrigated areas (Moore and Rojstaczer, 2002; DeAngelis et al., 2010; Carrillo-Guerrero et al., 2013; Harding et al., 2013). It can also affect the dynamics of the monsoon (Douglas et al., 2006; Saeed et al., 2009; Shukla et al., 2014) and influence climate at both regional and global scales (Sacks et al., 2009; Puma and Cook, 2010). These findings show a gradual and significant influence of changes in irrigated areas on the hydrological cycle (e.g. Adegoke et al., 2003; Haddeland et al., 2006; Rost et al., 2008;

Döll et al., 2009; Hanasaki et al., 2010; Biemans et al., 2011). The ability of numerical models to reproduce these different impacts and feedbacks is thus essential in order to understand the role of irrigation in the Earth climate system at different spatial scales (Zaitchik et al., 2005). Representing irrigation could potentially improve weather and climate forecast skill (Ozdogan et al., 2010). However, as presented below, irrigation is generally represented in models in a too simplistic way.

Land surface models (LSMs) represent land surface biophysical processes and variables, including soil moisture and vegetation biomass, in a way that is fully consistent with the representation of carbon, water and energy fluxes. However, current models have to improve the representation of anthropogenic factors and their interactions with natural processes (Verburg et al., 2016). In particular, LSMs need to represent the complexity of irrigation practices as much as possible, and their impact on

the environment. Efforts are made to achieve this goal in the Community Land Model (CLM) and Noah-MP LSMs (Felfelani et al. 2020, Zhang et al. 2020, respectively). However, as highlighted by Chukalla et al. (2015), many large scale LSMs currently represent only one type of irrigated vegetation (mostly C4 crops, i.e. crops with a C4 photosynthesis carbon fixation type, such as corn, sorghum), with only one type of irrigation practice (e.g. sprinkling or flooding), one season per year and no inter-annual

variability of vegetation density (Perry, 2007; Perry et al., 2009). Among others, this is the case in the current version of the ISBA (Interaction between Soil, Biosphere, and Atmosphere; Noilhan and Planton, 1989) LSM, with C4 crops irrigated with sprinkling (Voirin-Morel, 2003; Calvet et al., 2008). In reality, there are a lot of different vegetation types which can be irrigated, from orchards to pastures (FAO, 2014), and different irrigation techniques with different ways to apply water (above the

vegetation or directly on the ground for sprinkling and flooding irrigation techniques, respectively). Different irrigation types vary in (1) irrigation efficiency (Evans and Sadler, 2008; Jägermeyr et al., 2015), (2) the amount of freshwater used for irrigation per surface unit (FAO, 2014), and (3) impact on water resources (Khan and Abbas, 2007). Moreover, some specificities of irrigation such as the timing and frequency of water application can affect the ecosystem and atmospheric responses to irrigation

(Sorooshian et al., 2012). Some models include a representation of irrigation without having an interactive vegetation scheme and using climatological values instead (such as with the LIS-Noah

model, a NASA land information system and LSM combination, used in Lawston et al., 2015), thereby precluding inter-annual variability of vegetation density and the impact of irrigation on vegetation growth. Having a more complete irrigation description is needed to reproduce the irrigation seasonality, and to represent possible changes in crop phenology such as emergence and harvest dates. The impact of changing irrigation characteristics in a context of climate change could thereby be evaluated, such as increasing irrigation efficiency (currently around 56%; FAO, 2014) and freshwater saving potential (Perry et al., 2017; Koech and Langat, 2018).

The objective of this work is to develop and evaluate a more detailed representation of irrigation practices into the ISBA LSM within the SURFEX (SURFace EXternalisée) modelling platform (Masson et al., 2013).

Section 2 presents a description of the ISBA LSM, the new crop and irrigation scheme, the validation protocol, followed by a description of the observational datasets. Section 3 illustrates the impact of the new scheme when compared to simulations without crop phenology and without irrigation. An evaluation of the performance of the model is made over Nebraska. Section 4 discusses the added value and the limits of the newly implemented irrigation scheme. Finally, section 5 presents the conclusions and future research directions.

## 2 Materials and Methods

### 2.1 The ISBA land surface model

The ISBA model (originally described in Noilhan and Planton, 1989) is the LSM developed by the research department of Météo-France (Centre National de Recherches Météorologiques, CNRM). It is embedded into the SURFEX modelling platform (Masson et al., 2013; Voldoire et al., 2017; Le Moigne et al., 2018), and can provide initial land surface conditions to various atmospheric models (e.g. ALADIN in Fischer et al., 2005), or be forced by atmospheric conditions in offline (i.e. stand-alone) mode. SURFEX integrates different models describing ocean and terrestrial surfaces. Over land, specific models are used to represent water bodies, cities, and the soil-plant system. The latter is modelled by the ISBA LSM. In SURFEX, land cover is described by ECOCLIMAP-II (Faroux et al.,

2013). This study takes advantage of the ECOCLIMAP-SG (Calvet and Champeaux, 2020; Supplement S1) major update of ECOCLIMAP-II. The ISBA model can be coupled to the CTRIP model (Decharme et al., 2019, Munier and Decharme, 2021) which is specifically designed to represent water dynamics within rivers and aquifers. The SURFEX framework allows the coupling of terrestrial processes with atmospheric models and hydrological models. For agricultural drought and water resource monitoring, SURFEX can also be operated offline, forced by a pre-existing dataset of atmospheric variables. Only offline ISBA simulations are considered in this study. In SURFEX, the evolution of land surface states (surface temperature, albedo, roughness…) and fluxes (evaporation, sensible heat flux, ground heat flux, net ecosystem exchange of $CO_2$) is simulated for four different tiles: natural and cultivated lands (e.g. deciduous and broadleaf forests, tropical, temperate and boreal grasslands, crops, deserts, …), urban areas, oceans and inland waters (such as lakes). The ISBA LSM is used to simulate natural and cultivated lands.

In this study, the version of ISBA including photosynthesis and temporal dynamical LAI evolution in response to environmental conditions is used (ISBA-A-gs; Calvet et al., 1998; Gibelin et al., 2006), together with the multi-layer soil hydrology scheme described in Decharme et al. (2019). Phenology is entirely driven by photosynthesis and no growing degree-day model is used. The only phenology parameter is a minimum LAI value, of 0.3 $m^2m^{-2}$ for low vegetation. The SURFEX v8.1 version (Le Moigne et al., 2018) is used to do the simulations. Since this study focuses on irrigation, only the tile of natural and cultivated lands is simulated with ISBA, representing the evolution of soil (temperature and water profiles), vegetation (leaf-level and canopy-level photosynthesis, biomass, LAI and carbon fluxes), surface hydrology (runoff and drainage) and snow conditions. To represent the global-scale diversity of continental natural surfaces, twenty different surface types (hereafter referred to as "nature types") can be used in ECOCLIMAP-SG (see Fig. S1.1 and Table S1.2).

## 2.2 Irrigation modelling concept

In this study, a pre-existing simple irrigation scheme (Calvet et al., 2008) within the ISBA LSM is upgraded to build a new version able to work at a global scale and to represent several types of irrigation practices. The irrigation can be activated for ISBA versions able to simulate interactive

vegetation biomass and LAI. Sprinkler irrigation is represented by imposing an additional water flux forcing to the soil-plant system. Water is applied at a given time and over a certain period of time. A number of irrigation parameters need to be assigned such as the irrigation amount, the irrigation interval, the irrigation start and end times. A parsimonious approach is used in order to limit the number of parameters of the model. Table 1 lists the parameters and the values used by default in the model.

Using these values allows the model to predict a realistic amount of irrigation water over irrigated corn in southern France (Bonnemort et al., 1996; Voirin-Morel, 2003; Calvet et al., 2008). Irrigation is triggered using thresholds of the simulated extractable soil moisture content, when vegetation growth is limited by a soil moisture deficit. The plant water stress level is evaluated using a unitless soil wetness index along the root profile ($SWI_{root\_zone}$). A $SWI_{root\_zone}$ value close to one corresponds to a well-

watered soil, while a value close to zero indicates extreme stress. In order to trigger irrigation, the $SWI_{root\_zone}$ value is compared to predefined SWI thresholds given as input parameters. These SWI thresholds are evolving during the irrigation season and default values are fixed to 0.7 for the first irrigation, 0.55 for the second irrigation, 0.4 for the third irrigation, and 0.25 afterwards. This irrigation strategy tends to limit water applications when the plant is able to extract water from the soil. When a

SWI threshold is reached, irrigation is triggered with a predefined quantity of water of 30 mm (by default), following Calvet et al. (2008). The yearly sum of this irrigated water can be compared to the USGS data described in Section 2.4.3. The irrigation water flux is evenly distributed over a period of time of 8 hours (by default) and is applied on top of the vegetation canopy like precipitation. The irrigation water can be intercepted by vegetation canopy.

Moreover, specific crop phenology parameters such as emergence and harvest dates are used for irrigated crops. In practice, two dates are prescribed: emergence and harvest. This is a simple way to represent specific crop phenology attributes of irrigated crops. Between these two dates, irrigation is possible. Before the emergence and after the harvest, LAI is fixed at the model's minimum value (LAI = 0.3 $m^2$ $m^{-2}$).

## 2.2.1 New irrigation processes

In Lawston et al. (2015), three irrigation types are considered: sprinkler irrigation, flood irrigation and drip irrigation. In the new version of ISBA the same irrigation types are represented but a different modelling approach is used. In this study, the sprinkler irrigation type is used and evaluated. Flood and drip irrigation will be considered in a future work. The new crop and irrigation algorithm is based on several steps described below and in Fig. 2.

Firstly, the model determines whether fields within the grid cell can be irrigated, i.e. they are equipped for irrigation (e.g. water supply, valves, pipes…). This information is given by the irrigation map described in section 2.4.1.

Secondly, the model checks that the vegetation growth stage is compatible with irrigation. For crops, irrigation can be triggered after the emergence and until a few days before the harvest (by default two weeks). The new crop and irrigation scheme provides the option to support up to three plant growth seasons per year. The crop phenology parameters are not applied to wooded vegetation (trees and shrubs), and can be applied without irrigation. Irrigation can optionally be triggered without considering any specific crop phenology parameter but this option is not considered in this study.

The availability of resources (equipment or local water distribution) is taken into account through a default minimum time gap between two successive irrigations (Zhang et al. 2019). This default irrigation interval parameter value is a constant (7 days by default) but maps of irrigation intervals could be used when available.

Since a multi-layer soil hydrology scheme is used in the new irrigation model, the root-zone SWI ($SWI_{root\_zone}$) is a weighted average SWI value based on the soil volumetric water content profile ($Wc_i$, m$^3$ m$^{-3}$), the field capacity volumetric water content profile ($Wfc_i$, m$^3$ m$^{-3}$) and the wilting point profile ($Wwilt_i$, depending on clay and sand fraction, m$^3$ m$^{-3}$), for each soil layer $i$. The root fraction inside each soil layer ($f_{root_i}$) is used as a weighting factor:

$$SWI_{root\_zone} = \sum_{i=1}^{n_{soil}} f_{root_i} \times \frac{Wc_i - Wwilt_i}{Wfc_i - Wwilt_i} \tag{1}$$

where $n_{soil}$ is the total number of soil layers in the root zone. This value depends on the considered vegetation type. For example, $n_{soil} = 9$ for crops, with a rooting depth of 1.5 m.

In addition to sprinkler irrigation, the new model is able to represent drip or flood irrigation. In this case, the water flux is applied directly to the soil surface, with no leaf interception. Considering the static equipment used for drip irrigation, there is no irrigation interval ($\Delta t_{Wn} = 0$ day). In this study, only

sprinkling irrigation is considered as this is the dominant irrigation type in Nebraska. Drip and flood irrigation will be evaluated in future works. The activation of a given irrigation method is described in Supplement S5. Irrigation simulations are illustrated in Supplements S2 and S3 over southwestern France and over the Hampton irrigated area in Nebraska (Fig. 1e and Figs. S1.2 and S1.3), respectively. Observed monthly precipitation in Nebraska is presented for contrasting years in Supplement S4.

All the values of the model parameters in Table 1 have been set within a default configuration. These values can be user-defined for each nature type and for each grid cell, including, when possible, seasonal variations. See Supplement S5 for configuration details and possibilities.

### 2.2.2 New aggregation rules of irrigated and rainfed vegetation

The new crop and irrigation scheme is operated using ECOCLIMAP-SG (see Supplement S1). The best

achievable spatial resolution of ECOCLIMAP-SG is 300 m × 300 m. In contrast to previous versions of ISBA, there is no specific irrigated nature type in the new ECOCLIMAP-SG vegetation description. On the other hand, irrigation of all the nature types listed in Table S1.2 is possible. By default, six vegetation types are considered (three crop and three woody vegetation types as shown in Fig. S1.1). The new crop and irrigation scheme is able to represent the sub-grid heterogeneity of the irrigation

fractional coverage. This is illustrated over North America by Figs. S1.2 and S1.3, showing the fraction of irrigated C4 and C3 crops, respectively.   For each nature type, an irrigated and a non-irrigated fraction are considered at the simulation resolution. In order to prevent an excessive increase in the number of simulated nature types (potentially 20 non-irrigated and 20 irrigated times 3 irrigation types, i.e. a total of 120 types), involving a large increase of complexity, memory and computing cost, some

choices are made for the implementation:

1.    Selection of a limited number of irrigated nature types. The default implementation consists in six irrigated nature types. Temperate deciduous and evergreen trees types (No 8 and 10 in Table S1.2, respectively) can be used to represent fruits trees or olive trees for example,

respectively. Shrub type (No 15) can be used to represent, among others, vine plants, and

types No 19, 20 and 21 may represent irrigated crops (e.g. wheat, soybean, and corn, respectively).

    2.      Selection of the main irrigation method used for each grid cell and nature type, considering that in one grid cell there is only one dominant method for a given nature type (e.g. flooded rice in China or sprinkled corn in France).

Finally, the system state variables (soil water content, surface and soil temperature, vegetation biomass, etc.) differ in irrigated and non-irrigated parts of the cell. This implies to (1) duplicate a nature type if it is partially irrigated, (2) attribute for each grid cell the corresponding irrigated fraction, and (3) select the irrigation type for the irrigated fraction. Lastly, the two irrigated and non-irrigated nature types are treated separately but the same rooting depth and secondary parameters (see Table S1.1) are

used.

In order to limit the computing time, vegetation types can (optionally) be gathered. In this case vegetation "patches" are created. In ISBA, patch aggregation (Masson et al., 2013) is a method used to reduce the number of simulated nature types. It is based on the aggregation of the fractions of nature types, as shown in Fig. 3. The model primary parameters such as rooting depth, LAI and tree height are

weighted using the fractional coverage of each nature type in the grid cell. The mean parameter values are calculated following different laws: dominant, arithmetic averaging, inverse averaging or inverse of square logarithm averaging, depending on the considered parameters, as described in Noilhan and Lacarrère (1995) and Noilhan et al. (1997). In practice, the nature types to be aggregated (see the list in Fig. S1.1) within a grid cell are first chosen. Then, during the simulation, the fractions of nature type

composing each patch are added together (step 1 in Fig. 3) for each grid cell. The different primary parameters (trees height, LAI, …) are weighted by patch following the respective vegetation fractions (step 2 in Fig. 3). For secondary parameters (e.g. photosynthesis parameters in this study) a minimum number of patches is needed in order to avoid combining incompatible vegetation types (e.g. C3 crops and C4 crops).

In a first step (step 0 in Fig. 3) the differentiation between irrigated and rainfed nature types is done by computing the irrigated (and rainfed) fraction for each nature type and for each grid cell. Arithmetic averaging is used to cross information from the nature type fractional coverage and from the global irrigation fraction map described in Section 2.4.1. The ECOCLIMAP-SG land cover classification uses this additional data layer to compute the fraction of irrigated vegetation at the spatial

resolution of the model simulations. Nature types considered as irrigated (by default 6) are duplicated (meaning that for each of them a new nature type is created with the same parameters). This ensures the distinction of irrigated and rainfed soil water budget types. Then, as before, the nature types are aggregated by patch and the primary parameter values are computed (step 1 and step 2 in Fig. 3, respectively).

This change of the code structure based on the aggregation tool is a way to (1) maintain the continuity with previous versions of the code, (2) ensure flexibility for the number of irrigated nature types to be considered and (3) simulate distinct irrigated and rainfed fractions of a nature type.

    The new irrigation module represents water demand for irrigation, only, and irrigation is not explicitly limited by the lack of water resources. This has consequences on water conservation. Water

used for irrigation is usually withdrawn from aquifers, rivers or reservoirs. These compartments are not represented in ISBA but a new module dedicated to dam/reservoirs is currently under development.

## 2.3 Experimental design

The simulations and the evaluation of the new irrigation scheme are made over the state of Nebraska (United States of America, USA). This area presents a high density of irrigated fields (Fig. 1) and large

freely available observational datasets for evaluation. In this area, most irrigated field consist of corn (Zhang et al., 2020). In particular, we focus on a region where the irrigation is prominent: the south of the state of Nebraska (100-97°W, 40.25-41.25°N, Fig. 1e). The objective of the model evaluation is to demonstrate that the model is able to reproduce irrigation activities and that the irrigation scheme improves vegetation modelling and the associated surface fluxes as compared to observations.

The ISBA LSM simulations are made at a spatial resolution of $0.25° \times 0.25°$, over a 40-year period from 1979 to 2018. The initial values of the soil moisture and soil temperature profiles are

derived from a 20-year spin-up simulation by repeating year 1979. The same initial conditions are used for all the simulations, with and without crop and irrigation modelling. To evaluate the impact of irrigation, these simulations are run using the ECOCLIMAP-SG land cover classification within SURFEX (see Supplement 1). All nature types are grouped into 15 patches including three irrigated ones: shrubs (orchards), C3 crops (typically wheat and rice) and C4 crops (corn). This study focuses on the results of these last two nature types because there are hardly any irrigated orchards in Nebraska in the irrigation map described in Section 2.4.1. The dates of the irrigation season for corn are chosen in accordance with the literature (USDA and NASS, 2010) from May (emergence) to September (harvest), with a random picking of the day within those specific months. Three types of simulations are performed (Table 2): "ISBA_ref" without irrigation nor crop phenology (the benchmark), "ISBA_pheno" with only crop phenology attributes (emergence and harvest dates) and the complete "ISBA_pheno_irr" simulation with irrigation and crop phenology attributes. For the intercomparison of the simulations we select areas where the irrigation fractional coverage is larger than 50 % as determined from the irrigation map, in order to better assess the local effects of irrigation in offline simulations.

The reference ISBA_ref LAI simulations are compared with those from ISBA_pheno and ISBA_pheno_irr experiments, and with the 0.01° × 0.01° LAI satellite observations over areas in Nebraska where the vegetation is considered as C3 or C4 irrigated crops by ECOCLIMAP-SG. In addition to LAI, other variables are considered: gross primary production, evapotranspiration and land surface temperature. In order to compare the time series simulations with observations, the Pearson correlation coefficient ($r$) and the root-mean-square difference (RMSD) scores are used. For water and carbon fluxes, they are calculated using daily values.

$$r = \frac{\sum_{i=1}^{N}(y_i - \bar{y})(x_i - \bar{x})}{\sqrt{\sum_{i=1}^{N}(y_i - \bar{y})^2 \sum_{i=1}^{N}(x_i - \bar{x})^2}} \tag{2}$$

where $y$ and $x$ stand for observations and model simulations, respectively, and

$$\bar{y} = \frac{1}{N}\sum_{i=1}^{N} y_i, \quad \bar{x} = \frac{1}{N}\sum_{i=1}^{N} x_i \tag{3}$$

are observation and model simulation means, respectively.

$$RMSD = \sqrt{\frac{1}{N}\sum_{i=1}^{N}(y_i - x_i)^2}$$ (4).

$N$ represents the number of observations interpolated or aggregated to the model grid. $N$ is equal to the number of model grid-cells used in the calculation of the scores.

## 2.4 Data

### 2.4.1 Irrigation map

One of the main challenges of this study is to obtain an upgraded map of irrigation at the global scale, to be consistent with the resolution (300 m × 300 m) of the European Space Agency - Climate Change Initiative (ESA-CCI) land cover map used in ECOCLIMAP-SG. The 1 km × 1 km resolution global irrigation map proposed by Meier et al. (2018), based on a statistical approach and satellite data, is used. A reason to choose this product is that its development process is based (amongst other) on the ESA-CCI land cover product (v1.6.1), the same as the one used to develop the ECOCLIMAP-SG vegetation map (Supplement S1).

In order to transfer the Meier irrigation map (1 km × 1 km) to ECOCLIMAP-SG (300 m × 300 m), a spatial resampling of the Meier map is performed (https://doi.org/10.5281/zenodo.6011618). A simple majority rule is used by assigning to each 300 m × 300 m grid point of ECOCLIMAP-SG the irrigation status (irrigated or rainfed) of the main corresponding grid-cell of the Meier 1 km × 1 km map. An irrigation map at a spatial resolution of 300 m × 300 m is obtained, with a single vegetation type attributed to each grid cell together with the irrigation status. The main limitation of this map is that there is no information on the type of irrigation. In this study, we consider that all irrigation is of "sprinkler" type as this is the most common irrigation type in the USA and in Nebraska (AQUASTAT and FAO, 2019), where the testbed area of this study is located. This entails that irrigation water is added to the precipitation forcing over the irrigated agricultural parcels.

### 2.4.2 Atmospheric forcing

The simulations presented in this study are not coupled with the atmosphere. They are forced by a simulated atmospheric dataset of the European Centre for Medium-Range Weather Forecasts

(ECMWF): the ERA-5 atmospheric reanalysis at 0.25° × 0.25° (Hersbach et al., 2020). This global dataset was successfully used to force the ISBA LSM in previous studies (e.g. Albergel et al., 2019, Bonan et al., 2020). Beck et al. (2019) show that the ERA-5 precipitation dataset is reasonably consistent with gauge-radar data over CONUS, except for mountainous areas. A subset of the ERA-5 forcing over Nebraska is used for the time period from 1979 to 2018. This period is chosen in order to encompass various validation datasets. The following atmospheric variables are used to force the ISBA LSM and are taken from ERA-5 at an hourly time step: air temperature, wind speed, air specific humidity, atmospheric pressure, shortwave and longwave downwelling radiation and precipitation (liquid and solid).

### 2.4.3 Validation datasets

Six observation datasets are used (Table 3) to evaluate the simulations over Nebraska: the water used for irrigation, satellite-derived Leaf Area Index (LAI), gross primary production (GPP), evapotranspiration, land surface temperature (LST), and precipitation.

Precipitation data from the Grand Island and Lincoln weather stations (40.93°N – 98.76°W, 40.83°N – 96.76°W, "Gi" dot in Fig. 1e and "Li" dot in Fig. 1b, respectively) are used to evaluate the ERA5 precipitation forcing over Nebraska. The two weather stations are within 170 km of each other and correspond to contrasting environmental conditions. While the Grand Island station is located within a densely irrigated area, the Lincoln station is located at the Lincoln airport, which is surrounded by rainfed agricultural fields.

The water use records are provided by the US Geological Survey (USGS) through the National Water Information System (available at https://waterdata.usgs.gov/ne/nwis/wu, last access February 2022). Every 5 years from 1985 onward, the annual raw amount of water collected for irrigation is available by county together with conveyance loss and with the surface area of the irrigated vegetation. This allows us to compute the amount of water used for irrigation per unit surface area (in mm) over the specific studied zone in Nebraska (Fig. 1e). The USGS data we use cover the 1985-2019 time period. Because conveyance loss data are not available for 1995, this year is not taken into account. In order to assess the consistency of the simulated irrigation process with observations, the simulated number of

yearly irrigation events on irrigated areas in Nebraska is compared with the USGS irrigation water amount estimates. Irrigation water amount is obtained from the simulated number of irrigation events using the model default irrigation water amount of 30 mm per irrigation event. Only values of the mean and standard deviation of the yearly irrigation amount are compared. The comparison is made for the irrigated croplands (either C3 or C4 crops) as defined using the irrigation map (Section 2.4.1) within the

studied irrigated area in Nebraska (Fig. 1e).

The simulated LAI is compared with a satellite-derived LAI product at $0.01° \times 0.01°$ spatial resolution derived from SPOT-VGT and PROBA-V satellite data (up to May 2014, and after May 2014, respectively) by the European Copernicus Global Land Service (CGLS). This LAI product is described in Baret et al. (2013). We use Version 2 of this product (GEOV2). It is available every 10 days from

1999 onward. It does not cover the whole simulation time period (1979 to 2018).

The simulated GPP is compared to an upscaled estimate of GPP available at 0.5° from 1980 to 2013, from the FLUXCOM project (Jung et al., 2017). Al-Yaari et al. (2021) show that the FLUXCOM daily evapotranspiration product can be used as a benchmark over irrigated areas. Since evapotranspiration and GPP fluxes are closely connected to each other, it can be assumed that the

FLUXCOM GPP product is also sensitive to irrigation. The FLUXCOM product is based on a global machine learning model that does not have to be locally trained. However, it seems that three flux stations in Nebraska are used in the training as their data are included in the La Thuile dataset used to build FLUXCOM (Tramontana et al. 2016). These stations are located at 45 km at the north-east of the Lincoln weather station (e.g. Suyker and Verma, 2009), in a region where irrigation is present but not

dominant.

The simulated evapotranspiration is compared to the GLEAM satellite-driven model estimates of land evapotranspiration available from 2003 to 2018 (version v3.2b, Martens et al., 2017). The GLEAM data come at the same $0.25° \times 0.25°$ model's grid.

The simulated LST at 12h00 (local solar time) is compared to the LST derived from

geostationary meteorological satellites by CGLS at 12h00 (local solar time). This product has a spatial resolution of $0.05° \times 0.05°$ and is available from 2009 to 2018 (Freitas et al., 2013). It must be noticed

that in the version of the model used in this study, a single composite soil-vegetation energy budget is used and the thermal effect of crop residues is not represented. This means that over croplands, the simulated LST can differ from the vegetation temperature as seen from space.

In addition to the validation datasets, corn LAI observations at the field scale for various agricultural management conditions are available in Boedhram et al. (2001).

## 3 Results

The results presented below are focused on the impacts of the crop phenology and irrigation implementation on the simulated land surface variables over Nebraska. In addition to these results,
illustrations of the response to irrigation of simulated key land surface variables (SWI, LAI, GPP, evapotranspiration, LST) are shown over southwestern France and over the Hampton area in Nebraska in Supplements S2 and S3, respectively. In the case of Hampton, it can be noticed that the simulated irrigation mainly occurs in July and August (Fig. S3.1).

### 3.1 Irrigation: water use

In Fig. 4, the mean yearly irrigation amount for C3 and C4 crops for the ISBA_pheno_irr experiment is compared to the values derived from the observations from the USGS. The simulated irrigation amount presents a large interannual variability, with a minimum of 60 mm in 1993 and a maximum close to  390 mm in 2002. It must be noticed that 1993 is one of the wettest year recorded at the Lincoln weather station (https://lincolnweather.unl.edu/records/annual.asp, last access May 2022). The mean
simulated value of the yearly irrigation water amount used for irrigation ($271\pm75$ mm year$^{-1}$) slightly overestimates the observed one ($264\pm65$ mm year$^{-1}$), with a difference of +2.7%. This difference could be explained by the availability of the water resource, not explicitly accounted for by the model yet. The large observed irrigation amounts in 2000 and 2005, larger than 300 mm year$^{-1}$, are relatively well represented by the model. On the other hand, the observed small irrigation amount for the 2010 wet
year, is overestimated by about 110 mm year$^{-1}$.

## 3.2 Irrigation: plant growth

Figure 5 illustrates the mean seasonal and interannual variability of LAI in the most densely irrigated part of Nebraska for areas with a fraction of irrigated crops larger than 50 % in Fig. 1e, from 1999 to 2018. Table 4 presents the peak LAI characteristics. While the satellite LAI observations present a peak at the end of July, the modelled LAI is plateauing in August (Fig. 5). The data from Boedhram et al. (2001) show that the modelled LAI plateau in August at LAI values of about 3.5 $m^2m^{-2}$ is realistic for irrigated corn. The satellite LAI observations are sensitive to both rainfed and irrigated vegetation. In all ISBA LAI simulations, the start of the growing season corresponds to a gradual increase in LAI from the initial value of LAI = 0.30 $m^2$ $m^{-2}$ imposed to the model in winter. The observed LAI presents a smaller minimum LAI value of 0.15 $m^2$ $m^{-2}$, starts increasing in April and a value of 0.30 $m^2$ $m^{-2}$ is reached at the end of April. Then, plant growth continues at about the same low rate till the end of May. The LAI growth rate increases in June and LAI reaches a mean peak value of 4.9 (±0.8) $m^2$ $m^{-2}$ is observed on 31 July (Table 4). The observed LAI then sharply decreases to reach its minimum value at about the end of September.

The ISBA_ref LAI simulations do not mirror the observed late growing season and rapid senescence. The ISBA_ref vs. observations comparison shows that without irrigation the simulated LAI generally starts increasing in March. On average, a peak LAI value of 3.6 (±0.2) $m^2$ $m^{-2}$ is simulated by ISBA_ref on 2 July, before slowly decreasing until the end of December. The ISBA_ref growing season is much longer than observed. It starts two months before the observations and stops three months after the observations. The simulated LAI peaks one month before the observations. The simulated yearly LAI amplitude is 28 % smaller than observed.

The ISBA_pheno LAI simulation is much more consistent with the LAI observations. The growing season starts in mid-May and the senescence ends at the end of September. However, the simulated peak LAI is still 30 % smaller than observed (LAI = 3.5 (±0.2) $m^2$ $m^{-2}$). The peak LAI is reached on 26 August, much later that the ISBA_ref peak LAI, and about one month after the observed peak. The sharp decrease of LAI in September results from harvests at random dates in September. Adding irrigation (ISBA_pheno_irr) does not change the general pattern of the LAI curve, but increases

the LAI amplitude, with a mean peak LAI value of 3.7 ($\pm$0.1) $m^2$ $m^{-2}$ on 28 August, larger (+8%) than for ISBA_pheno but still below the observation (-24%).

The interannual variability of simulated and observed LAI values is illustrated in Fig. 5b, from 2002 to 2008. The ISBA_ref LAI presents a systematic drop in summer, which is not present in the observations nor simulated by the ISBA_pheno and ISBA_pheno_irr experiments. Without the regular seasonality imposed by crop phenology parameters, the model may simulate a re-growth of vegetation in autumn (e.g. in 2003), that is not present in the observations. The ISBA_pheno and ISBA_pheno_irr

simulations are more consistent with the observed seasonality.

### 3.3 Impact of crop phenology and irrigation on LAI at a regional scale

This section is focused on the impact of irrigation practices for the south Nebraska zone (as defined in Fig. 1e), and all nature types are considered for the comparison with observations at a spatial resolution of $0.25° \times 0.25°$.

Figure 6a shows the seasonal mean LAI variations from 1999 to 2018. This Figure is similar to Fig. 5a but all nature types are considered. Peak LAI characteristics are given in Table 4. They differ from the crop LAI peaks. While, the observed LAI peaks at 3.8 ($\pm$1.5) $m^2$ $m^{-2}$ on 31 July for ISBA_ref, LAI peaks at 3.3 ($\pm$0.3) $m^2$ $m^{-2}$ on 1 July for ISBA_ref, 3.1 ($\pm$0.3) $m^2$ $m^{-2}$ on 16 July for ISBA_pheno, and 3.1 ($\pm$0.3) $m^2$ $m^{-2}$ on 16 July for ISBA_pheno_irr. Compared to crop simulations, the experiments

with crop phenology (ISBA_pheno and ISBA_pheno_irr) present earlier peak LAI dates, because rainfed vegetation affects the phenology. However, the irrigated crop signature is visible in the second peak of the annual LAI cycle simulated by ISBA_pheno and ISBA_pheno_irr experiments at the end of August. More often than not (83 % and 88 % of the grid cells for $r$ and RMSD, respectively) the LAI score differences between ISBA_pheno_irr and ISBA_ref shown in Fig. 6 correspond to an

improvement of the LAI simulation with the representation of irrigation. A month by month analysis of the scores (Fig. 7) shows a marked improvement of $r$ values in June, July and September. The $r$ value can frequently be increased by 30%. Lower RMSD values are observed from April to November, more frequently in May and in October. In April, October and November, the main cause of the reduction in RMSD values is the imposed minimum value of 0.3 $m^2$ $m^{-2}$ before the emergence (in May) and the

harvest (in September). The ISBA_pheno correlation and RMSD differences with respect to ISBA_ref are nearly identical to those showed for ISBA_pheno_irr in Figs. 6-7 (not shown).

**3.4 Impact on the GPP flux**

As the vegetation productivity is linked to LAI, the seasonality pattern of GPP (Fig. 8) is comparable to the one of LAI (Fig. 6) but the observed GPP peak ($9.2 \pm 2.1$ g[C].m$^{-2}$.day$^{-1}$) occurs on mid-July while

the observed LAI peaks on 31 July. During the plant growth period, the smallest differences between all the simulations and the observations occur at about the same time as the observed GPP peak. For all simulations, a GPP plateau at a value of $9.0 \pm 1.8$ g[C] m$^{-2}$ day$^{-1}$ is reached at the beginning of July and lasts until mid-July. Finally, the simulated GPP decreases in September with a delay of about two weeks with respect to the observations.

Before July, the ISBA_ref photosynthetic activity is well in advance as compared to the observations, of about 20 days in May. This is consistent with the very large LAI values simulated by ISBA_ref in May: about 2 m$^2$ m$^{-2}$, while the mean LAI observation hardly exceeds 0.5 m$^2$ m$^{-2}$. The simulated GPP maximum ($9.7 \pm 2.0$ g[C] m$^{-2}$ day$^{-1}$) is reached before the end of June. After a sharp decrease at the end of June, the ISBA_ref GPP decreases at a slower rate than the observations. From

mid-September to the end of October, the simulated GPP is much larger than the observed GPP.

The ISBA_ref flaws are much less pronounced in ISBA_pheno and ISBA_pheno_irr experiments. In the latter simulations, the increase of the GPP occurs at about the same time as in the observations. The GPP values are systematically larger with irrigation in July and August than for other simulations. As for LAI, the GPP *r* and RMSD scores (Fig. 8b and 8c, respectively) are better for

ISBA_pheno_irr than for ISBA_ref, with an improvement on 87 % and 81 % over the domain, respectively.

**3.5 Impact on evapotranspiration**

Investigating evapotranspiration is a way to assess the impact of irrigation on the hydrological system. Figure 9 shows evapotranspiration for the ISBA simulations and for GLEAM. This Figure is similar to

Figs. 6 and 8 but a shorter time period is considered, from 2003 to 2018. Before the irrigation period,

the observed evapotranspiration steadily increases from February to July. After the irrigation period, evapotranspiration decreases until November. It can be observed that the short term variability of the GLEAM evapotranspiration is represented well by the simulations. On the other hand, all ISBA simulations produce much larger evapotranspiration values than GLEAM during the growing period from April to June. For example, all ISBA simulations can reach 5 mm day$^{-1}$ while GLEAM does not exceed 3.5 mm day$^{-1}$. Over this time period, ISBA_ref overestimates evapotranspiration with respect to GLEAM by 0.98±0.42 mm day$^{-1}$ (38±16 %) on average.

On the contrary, from mid-July to mid-August, all ISBA simulations tend to underestimate evapotranspiration with respect to GLEAM, by up to 1.3 mm day$^{-1}$ for ISBA_ref. Accounting for crop phenology and irrigation into the model has a substantial impact on this variable and reduces the bias. Over the whole irrigation period, the mean bias goes from -0.4 ± 0.4 mm day$^{-1}$ (-13 ± 12 %) for ISBA_ref to -0.2 ± 0.3 mm day$^{-1}$ (-7 ± 11 %) and -0.1 ± 0.3 mm day$^{-1}$ (-2 ± 11 %) for ISBA_pheno and ISBA_pheno_irr, respectively. Evapotranspiration is overestimated again after the harvest, from mid-September to November by 0.38±0.18 mm day$^{-1}$ (42±20 % compared to the observations).

The newly implemented processes have a small but positive impact on the bias before and after the irrigation period. During the growing season, from April to June, the overestimation decreases from 38 % in ISBA_ref to 33 % and 34 % for ISBA_pheno and ISBA_pheno_irr, respectively. From mid-September to November, the overestimation decreases from 42 % to 35 % and 36 %, respectively. The *r* and RMSD differences between ISBA_ref and ISBA_pheno_irr (Fig. 9) also show a global improvement with 83 % and 79 % of the grid cells being improved. However, the effect on the *r* score is small (less than 0.1) and heterogeneous in time and space. Figure 10 shows that the *r* score is mainly improved in August and in September, before the harvest. Degradation in *r* can be observed at some locations throughout the growing season. The improvement of RMSD is more stable, and can be observed from May to October, the impact being more pronounced in July and August.

## 3.6 Impact on LST

In order to evaluate the impact of irrigation on the land surface energy budget, Fig. 11 shows land surface temperature at 12h00 local time simulated by the three model configurations and derived from

the CGLS product. Overall, ISBA tends to overestimate LST at noon, especially in April-May, up to 7 °C in Fig. 11a. The bias is reduced during the summer.

Due to the difficulty of observing the differences between the simulations, Fig. 11b presents differences of ISBA_pheno and ISBA_pheno_irr versus ISBA_ref. With crop phenology (with or without irrigation) the simulated LST is globally higher from April to June and from mid-September to November. The maximum difference with respect to ISBA_ref is +0.7±0.3 °C. It is observed for ISBA_pheno in September. During the summer (July and August) the new model versions tend to

present lower LST values, with temperature differences close to -0.2 ± 0.1 °C in ISBA_pheno_irr. Moreover, from May to mid-September the temperature in ISBA_pheno_irr is lower than in ISBA_pheno, and this difference can reach locally -0.9°C in summer. Figure 12 presents the monthly $r$ and RMSD scores of ISBA_ref with respect to CGLS LST observations and the ISBA_pheno_irr score difference is shown. It shows that there is a seasonal dependence of these statistical values, with slightly

better $r$ and RMSD scores observed for ISBA_pheno_irr in July and August during the irrigation period. However, the representation of irrigation tends to degrade RMSD before (April, May) and after (October, November) the irrigation period.

## 4 Discussion and perspectives

The results presented in Section 3 show that the new version of ISBA is able to produce a realistic

yearly irrigation water amount (Fig. 4). It also markedly improves the LAI and GPP simulations (Figs. 5-6 and Fig. 8, respectively). On the other hand, the new ISBA version developed in this study has a limited impact on the evapotranspiration and on the LST simulations and is not able to significantly reduce the strong model biases that are observed for these variables before and after the irrigation time period (Figs. 9-11).

**4.1 Could the new crop and irrigation scheme be further improved?**

The crop phenology model is very simple and adding more parameters related to phenology could be a way to further improve the model performance. Integrating satellite LAI observations in ISBA using

sequential data assimilation is also an option (Mucia et al., 2020). The results of our numerical experiments over Nebraska show that considering crop phenology improves the consistency of the simulations with LAI and GPP observations. The corresponding correlation and RMSD scores are improved. The crop phenology parameters used to force emergence and harvest dates reduce the length of the growing season, delay spring growth and avoid a regrowth in the autumn. It seems that irrigation only plays an additive role in improving the vegetation seasonal cycle as compared to the role of including crop phenology (Section 3.3). Both crop phenology and irrigation models have shortcomings and their performance could be limited by difficulties in simulating processes that are not directly related to irrigation.

Firstly, the same emergence and harvest dates are imposed for all years, while in reality crop phenology may present an inter-annual variability related to climate conditions. This is particularly the case for Nebraska because the start of the growing season depends on the snowmelt and soil thawing dates. These processes are represented in ISBA and crop phenology parameters could be related to snow melting and soil thawing, but this would require extensive developments to be implemented at a global scale. Moreover, the representation of the cold season processes is not perfect in ISBA (Decharme et al. 2019) and the model tends to underestimate snow depth and the length of the snow season. This could explain biases in soil temperature and LST simulations before and after the irrigation time period. Figure 11 shows that LST values below the freezing level can be observed in April and that their model counterparts are about 7 °C warmer. The earlier thawing in model simulations is reflected in the much earlier leaf onset in LAI simulations. Figure 6 shows that while the observed LAI does not exceed 0.5 $m^2$ $m^{-2}$ at the end of April, the ISBA_ref LAI reaches the same value about one month earlier. The unrealistically early leaf onset is consistent with the warm model bias at the end of the cold season. This shows that improving the representation of the cold season by assimilating satellite-derived or in situ snow cover fraction observations could improve the simulation of the crop growing period in this area. Also, emergence and harvest dates could be derived from the LAI observation in order to better represent the interannual variation. However, the currently used empirical approach to establish the crop season provides robust results over the irrigated grid cells (Fig. 5).

Secondly, the irrigation itself is based on fixed parameter values such as the minimum period between two consecutive irrigations (one week) and SWI levels triggering irrigation turns. The simulations over the Hampton grid cell show that the first irrigation can start at quite low levels of the SWI (Fig. S3.1), even below the second irrigation threshold of 0.55 defined in Section 2.2.1. Suppressing the one week constraint of irrigation turns improves the simulation of the peak LAI, which

otherwise is rather poorly simulated (Fig. 5). However, this change triggers unrealistic large irrigation water amounts (not shown). A lack of irrigation water amount cannot explain the excessive soil water deficit. One could also challenge the quality of the ERA5 precipitation. Figure S4.1 and Fig. S4.2 show that ERA5 precipitation compares well with in situ observations and that the seasonal and inter-annual variability is fairly represented. A more plausible explanation could be that the initial soil water storage

value between the end of the cold season and the first irrigation turn is withdrawn too quickly from the soil by the model. This explanation would be consistent with the marked overestimation of evapotranspiration in spring, from April to June (Fig. 9), before the irrigation time period.

A possible limitation of using a global low-resolution reanalysis such as ERA5 is that changes to the local climatic conditions caused by irrigation may not be represented. Over Nebraska, Szilagyi and

Franz (2020) show that the decadal increase in irrigated land tends to trigger a reduction in precipitation over the most densely irrigated areas, of about -10 mm per decade. The largest precipitation suppression is observed at Spring, in March, before the corn growing season, in relation to larger soil water content values. In our simulations, ISBA_pheno_irr presents larger soil moisture values than ISBA_ref in March (see Fig. S3.1), but this is mainly due to crop phenology. The ERA5 screen-level 2 m air temperature

and relative humidity are analyzed together with soil moisture by assimilating in situ observations from ground weather stations (Hersbach et al. 2020). In large irrigated areas where weather stations are present, the assimilation should be able to represent the soil moisture effect on these variables, even at coarse spatial resolution. A large-scale experiment involving ground and airborne measurements was recently performed in northeastern Spain to assess the impact of irrigation on atmospheric model

simulations (Boone et al. 2021).

## 4.2 Are evaporation components simulated well?

In order to investigate the evapotranspiration bias in Spring, the evaporation components are plotted in Fig. S3.3 and Fig. S3.4 for the Hampton irrigated area in 2018. Figure S3.3 shows that total evapotranspiration of ISBA_ref and ISBA_pheno_irr are quite similar. This is consistent with the small impact of crop phenology on total evapotranspiration showed in Fig. 9. On the other hand, soil evaporation and plant transpiration differ. In the ISBA_pheno_irr simulation, transpiration is reduced in Spring by more than 30 %, in comparison with ISBA_ref. The lower transpiration is offset by larger soil evaporation values. As a result, total evapotranspiration does not change much and the bias is not reduced in ISBA_pheno_irr. Also, Fig. S3.4 shows that the new crop and irrigation module does not affect interception much. Therefore, the ISBA_pheno_irr evapotranspiration bias in spring could be caused by the large soil evaporation. The evaporation component could be overestimated because (1) the soil is too warm in relation to a poor representation of thawing or because (2) crop residues at the soil surface are not represented. Wortmann et al. (2012) show that in this area, not harvesting crop residues tends to reduce soil evaporation and increase crop yield, limit water runoff, soil erosion, and contributes to maintaining soil fertility. Suyker and Verma (2009) show that increasing surface mulch dry mass from 50 to 150 g m$^{-2}$ can decrease the non-growing season evapotranspiration by more than 20 %. The ISBA model includes a representation of litter in forests (Napoly et al. 2017) that will be generalized to low vegetation in the next version of SURFEX. Using this new capability could improve our simulations.

Finally, degradation in *r* for evapotranspiration in Fig. 10 may be evidence that GLEAM may not be considered as a suitable reference for evapotranspiration comparisons in areas impacted by irrigation. The use of other datasets is investigated in Supplement 3. In particular, in situ observations over an irrigated corn field (Suyker and Verma, 2009) are used. Table S3.1 shows that GLEAM tends to underestimate evapotranspiration by 20 % during the growing season (from May to September). During the non-growing season, the ISBA_pheno_irr model overestimates evapotranspiration by 48 %. Table S3.2 and Fig. S3.5 show that the ISBA_pheno_irr evapotranspiration peak (in June) tends to happen too early. Mean values of near-surface wind speed are particularly large over Nebraska, especially at

wintertime and springtime (Chen, 2020). This feature could exacerbate the impact of a misrepresentation of soil evaporation.

## 4.3 Is the irrigation scheme flexible enough?

In this study, sprinkling irrigation is considered. The model is able to represent other irrigation systems such as flooding irrigation but more developments are needed to limit the runoff to the irrigated plot and this options needs to be validated. The newly implemented irrigation processes, along with the new ECOCLIMAP-SG vegetation description let users choose which nature type should be irrigated. Irrigation can be represented at various spatial scales, ranging from the field scale for agricultural studies to the global scale for climate studies. Model parameters can be specified using new datasets or local characteristics. For example, in this article we use a unique date for starting and ending the crop growing season with a random variability, but more accurate dates can be prescribed (varying spatially and from one vegetation type to another, or using crop calendars). Moreover, the better spatial resolution of ECOCLIMAP-SG allows the use of high resolution atmospheric forcing. This provides new opportunities for assessing the impact of irrigation on local climate and water resource conditions.

This study is mainly focused on a zone in the south of Nebraska where the irrigation density is relatively high (Fig. 1), and results could differ in other regions. Except for the fixed emergence and harvesting dates corresponding to regional crop phenology (from USDA and NASS, 2010), default values are used for all the other parameters (Section 2.4). Tests performed in southwestern France (Supplement S2) allow ensuring that the model is able to work in contrasting climate conditions.

In this study, the ISBA simulations are not coupled to the atmosphere, nor to the CTRIP river routing system. Such coupled numerical experiments can be performed thanks to the SURFEX modelling platform. However, more developments are needed in order to ensure water conservation in the hydrological system. In particular, irrigation water amounts should be consistent with the available water resource in rivers, groundwater, and dams.

## 5 Conclusions

A new uncalibrated irrigation scheme is implemented within the ISBA land surface model in order to improve the representation of vegetation over agricultural areas. A case study over an irrigated area in the state of Nebraska (USA) is performed to validate the new scheme. Simple crop phenology rules represent emergence and harvesting and improve the seasonality of plant growth, while the additional water supply from the irrigation mostly impacts the peak LAI value. The model is able to produce a realistic yearly irrigation water amount and markedly improves the LAI and GPP. It is shown that model performance can be limited by processes not directly related to irrigation, such as thawing or crop residues. The irrigation scheme has many possible configurations and the code is highly flexible. With this capability, ancillary data on farming practices such as emergence and harvest dates, or the amount of water per irrigation event, can be used. This flexible crop phenology and irrigation scheme can take the spatial heterogeneity of irrigation activities into account, and detect irrigation-induced impacts on Earth system simulations. Our results show that crop phenology parameters modify the seasonal pattern of the simulation of LAI, soil moisture, evapotranspiration and plant carbon uptake, and that irrigation affects their magnitude. This provides the basis for further development in offline and online applications of the ISBA model.

## Code availability

The ISBA land surface model is available as open source via the SURFEX modelling platform, available at https://www.umr-cnrm.fr/surfex/spip.php?article387. It is under a CECILL-C License (French equivalent to the L-GPL licence). The version developed and use for the experiment in this study is available on: https://doi.org/10.5281/zenodo.5718063. It based on the SURFEX version 8.1 (ref f70f6457). For future use, it is strongly recommended to use the newest version of ISBA, from the version 9.0 (scheduled for release in 2022) from which the irrigation developed will be included by default. Initialization files are available on: https://doi.org/10.5281/zenodo.6011618.

**Author contribution**

Arsène Druel and Clément Albergel designed the experiments. Arsène Druel carried out the
660 implementation of the irrigation scheme and performed the simulations. Arsène Druel wrote the
manuscript. All co-authors participated to the analysis of the results and to the revision of the
manuscript.

**Competing interests**

The authors declare that they have no conflicts of interest.

**Acknowledgments**

The work presented here was supported by the project URCLIM (advance on URban CLIMate services,
part of ERA4CS, an ERA-NET initialised by JPI Climate with co-funding of the European Union
(Grant n°690462)). The authors would like to thanks Stephanie Faroux and Marie Minvielle in charge
of the SURFEX code support for technical assistance, and Deborah Verfaillie for her careful reading of
670 the manuscript.

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

**Table 1 – Irrigation parameters.**

| Symbol | Definition | Default value (this study) |
|---|---|---|
| $I_T$ | Irrigation type | sprinkler |
| $I_{NT}$ | Irrigated nature type | C3 crops, C4 crops, shrubs |
| $I_W$ | Water amount per irrigation water turn | 30 mm |
| $I_D$ | Irrigation water turn duration | 8 hours |
| $SWI_1$ | SWI threshold for triggering the first water turn | 0.70 |
| $SWI_2$ | SWI threshold for triggering the second water turn | 0.55 |
| $SWI_3$ | SWI threshold for triggering the third water turn | 0.40 |
| $SWI_{4+i}$ | SWI threshold for triggering the following water turns (i, integer > 0) | 0.25 |
| $\Delta t_{Wn}$ | Minimum time lapse between two water turns (irrigation interval) | 7 days (0 days for drip irrigation) |
| $\Delta t_{WH}$ | Minimum time lapse between the last water turn and the harvest | 15 days |
| $t_E$ | Emergence date | 15 May ($\pm$ 15 days) |
| $t_H$ | Harvest date | 15 September ($\pm$ 15 days) |


**Table 2 – Main set up of the three 40-year evaluation experiments forced by ERA-5 atmospheric variables over Nebraska. Crop phenology is defined by emergence and harvest dates, while**
**irrigation corresponds to additional water supply.**

| Experiment | Crop phenology | Irrigation | Forcing | Spinup time | Simulation time period |
|---|---|---|---|---|---|
| ISBA_ref | no | no | | | |
| ISBA_pheno | YES | no | ERA-5 0.25° × 0.25° | 20 years | 1979-2018 |
| ISBA_pheno_irr | YES | YES | | | |


**Table 3 – Evaluation datasets.**

| Observations | Source | Reference | Spatial resolution | Time period | Sampling time |
|---|---|---|---|---|---|
| Water used for irrigation | USGS | https://waterdata.usgs.gov/ne/nwis/wu | County | 1985-2015 | 5 years |
| LAI | CGLS | Baret et al., 2013 | 0.01° | 1999-2018 | 10 days |
| GPP | FLUXCOM | Jung et al., 2017 | 0.25° | 1980-2013 | 1 day |
| Evapotranspiration | GLEAM | Martens et al., 2017 | 0.25° | 2003-2018 | 1 day |
| Land surface temperature at 12h | CGLS | Freitas et al., 2013 | 0.05° | 2009-2018 | 1 day |
| In situ precipitation | University of Nebraska-Lincoln | http://climod.unl.edu/ | local | 2009-2012 | Monthly |

**Table 4 – Observed and simulated mean LAI peak characteristics over Nebraska for the 1999-2018 time period for crops (see Fig. 5) and all vegetation types (see Fig. 6).**

| Vegetation types | LAI source | Peak LAI ($m^2$ $m^{-2}$) | Peak LAI date |
|---|---|---|---|
| Crops | Satellite observations | 4.9 (±0.8) | 31 July |
| | Boedhram et al. 2001 (*) | 3.6 to 4.0 | 12 July to 19 August 1994 |
| | Boedhram et al. 2001 (*) | 3.5 | 2 August to 23 August 1995 |
| | ISBA_ref | 3.6 (±0.2) | 2 July |
| | ISBA_pheno | 3.5 (±0.2) | 26 August |
| | ISBA_pheno_irr | 3.7 (±0.1) | 28 August |
| All | Satellite observations | 3.8 (±1.5) | 31 July |
| | ISBA_ref | 3.3 (±0.3) | 1 July |
| | ISBA_pheno | 3.1 (±0.3) | 16 July |
| | ISBA_pheno_irr | 3.1 (±0.3) | 16 July |

(*) Boedhram et al. (2001) data are for fertilized irrigated corn in 1994 and 1995


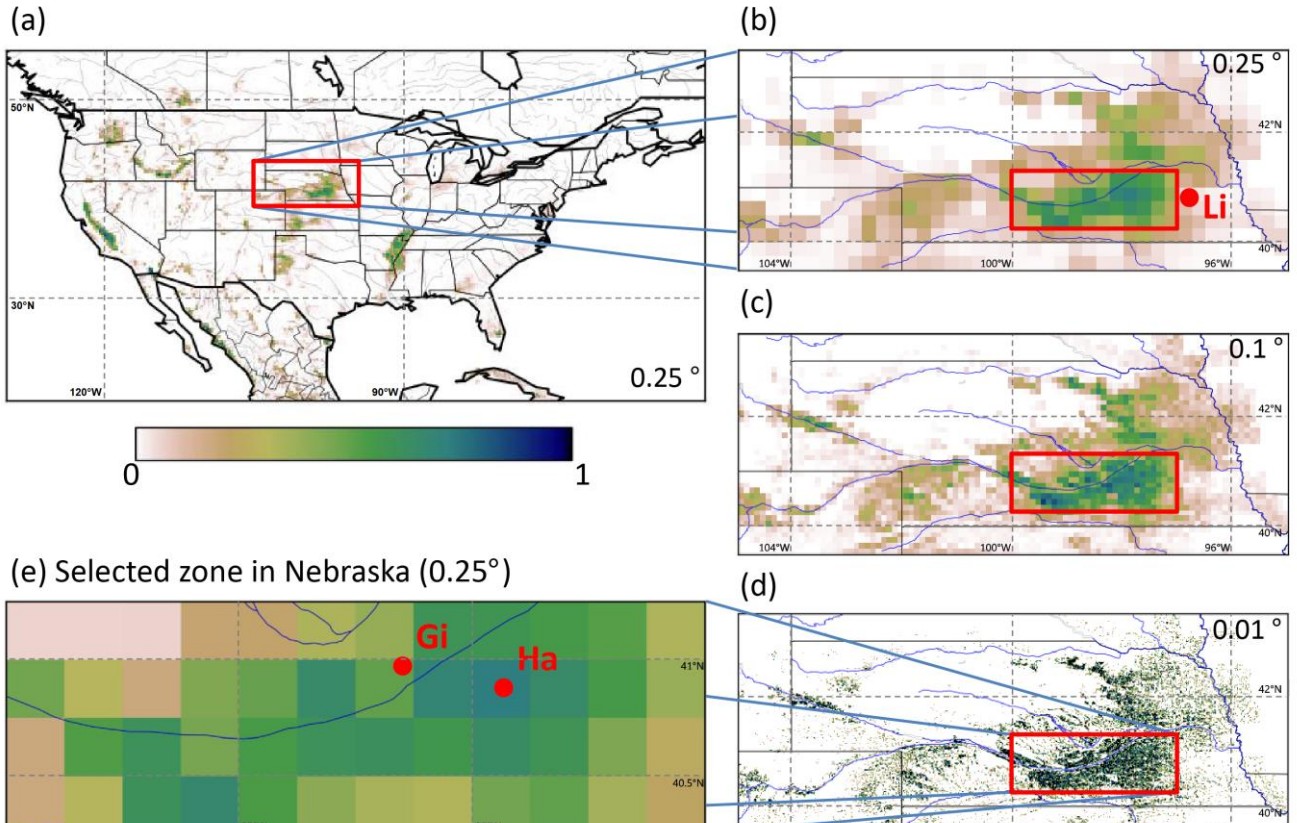

**Figure 1 – Irrigation fractional coverage derived from Meier et al. (2018) over (a) the Continental United State (CONUS), (b, c, d, e) Nebraska: at (b) 0.25°×0.25°, (c) 0.1°×0.1°, (d) 0.01°×0.01° spatial resolutions, and (e) over the selected zone in southern Nebraska considered in this study (100-97°W, 40.25-41.25°N). The red boxes show the location of the different zooms. The "Li", "Gi" and "Ha" red dots correspond to the Lincoln weather station, Grand Island weather station, and Hampton irrigated area, respectively.**

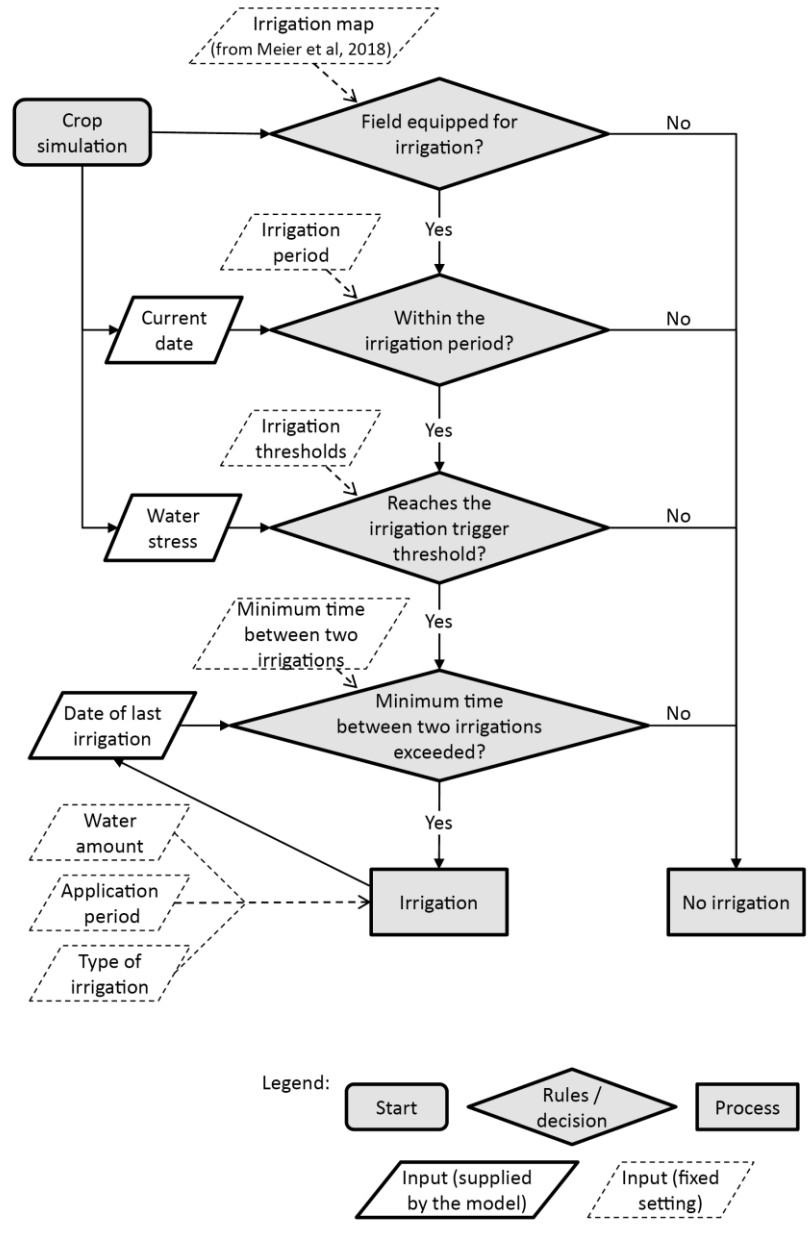


**Figure 2 – Irrigation decision tree model.**

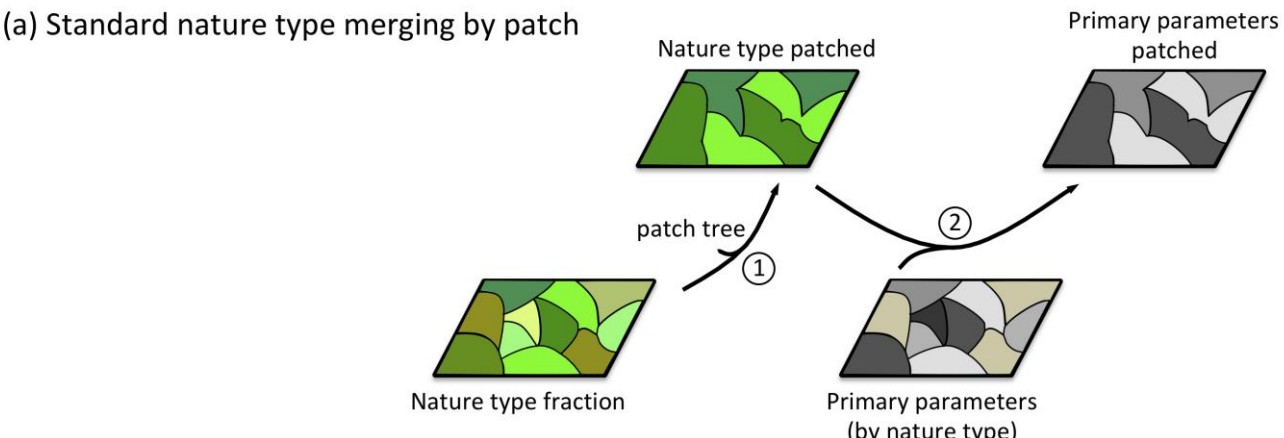


**Figure 3 – Diagram of the processing steps to obtain the ISBA model primary parameters from the nature types: (a) with the original method and (b) with the new method developed for irrigation. The different steps consist of: (0) cross-referencing the maps of vegetation cover (nature types) and irrigation fractional coverage (addition of irrigated nature type classes where**
**irrigation is possible), (1) merging nature type classes following path aggregation rules (see Fig. S1.1), and (2) computing primary parameter values following the patch faction map.**

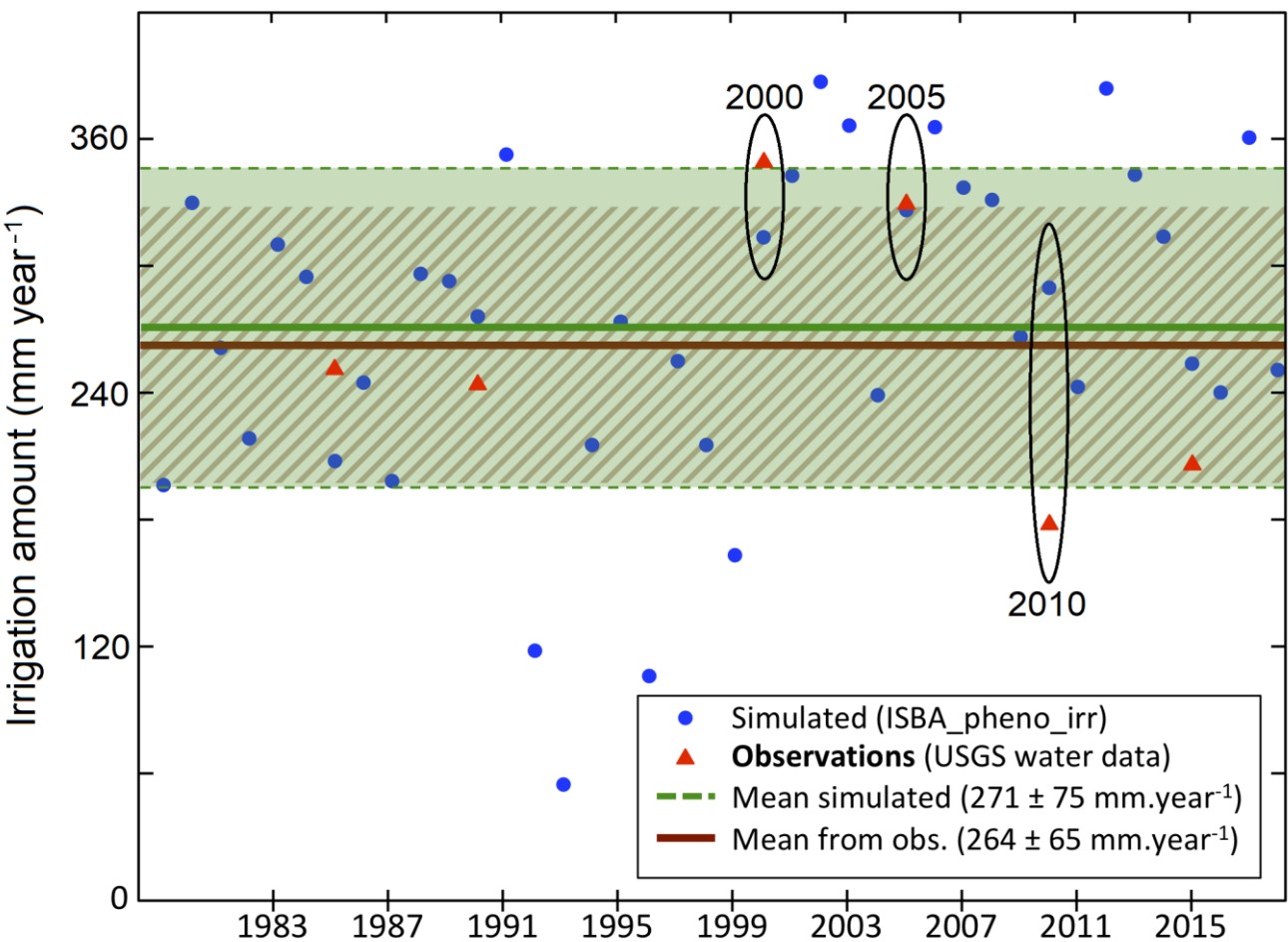

**Figure 4 – Yearly cumulated irrigation amounts simulated by the model for the studied area in Nebraska from 1979 to 2018 (blue line). The mean and standard deviation of the yearly values are shown for the model (green solid and dashed lines, respectively), and for the USGS water data from 1985 to 2019 (brown lines). The 2000 and 2005 dry years are indicated, together with the 2010 wet year.**

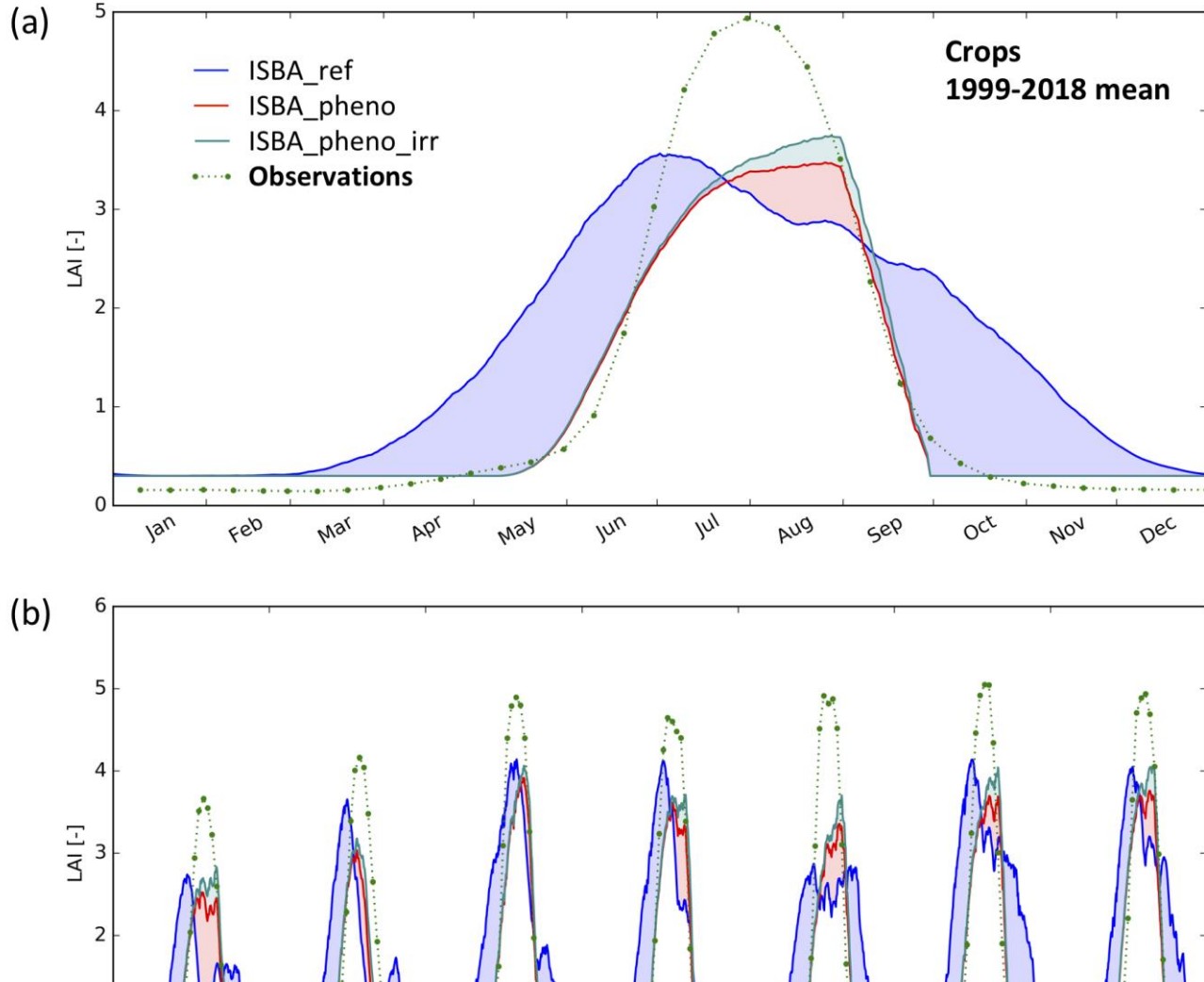


**Figure 5 – LAI (m² m⁻²) of irrigated crops (C3 or C4) in the most densely irrigated part of Nebraska (Fig. 1e): (a) seasonal variation for the time period from 1999 to 2018, (b) daily time series from 2002 to 2008. Simulated LAI is shown for the irrigated fraction, from the reference simulation (ISBA_ref, blue line), and from the simulations with only agricultural practices and**

**with agricultural practices and irrigation (ISBA_pheno, red line, and ISBA_pheno_irr, cyan line, respectively). Satellite-derived LAI observations (green dots) are for areas where the fraction of C3 or C4 irrigated crops is larger than 50 %.**

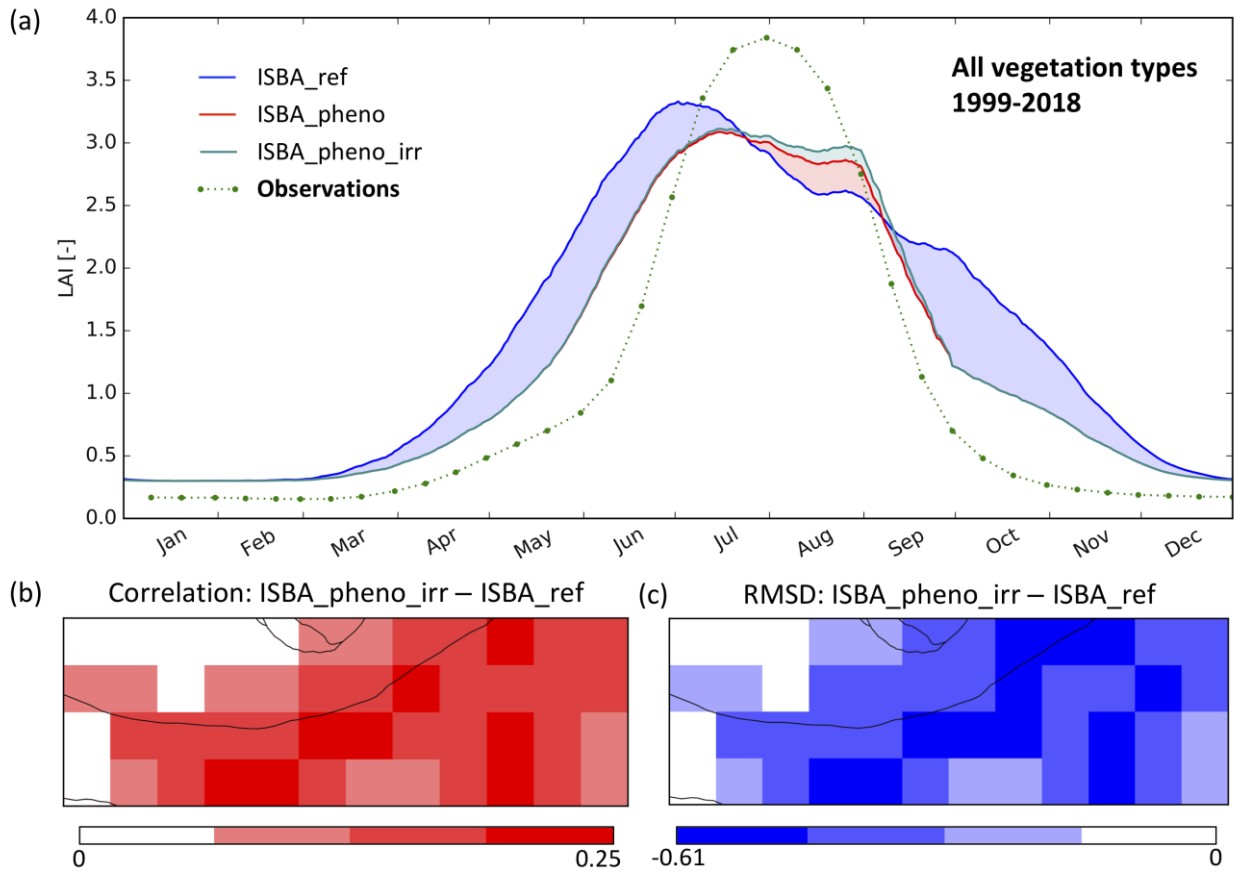

**Figure 6 – Simulated vs. observed LAI (m²m⁻²) of all vegetation types, in the most densely**
**irrigated part of Nebraska (Fig. 1e) from 1999 to 2018: (a) seasonal variation of mean LAI of**
**ISBA_ref (blue line), ISBA_pheno (red line), ISBA_pheno_irr (cyan) simulations and of satellite-**
**derived observations (green dots), (b) temporal correlation and (c) RMSD score difference maps**
**showing the added value of the ISBA_pheno_irr with respect to ISBA-ref. Positive value for**
**correlation (Fig. 6b) means that the result of ISBA_pheno_irr is better, and for RMSD (Fig. 6c)**
**negative value means that result of ISBA_pheno_irr is better.**

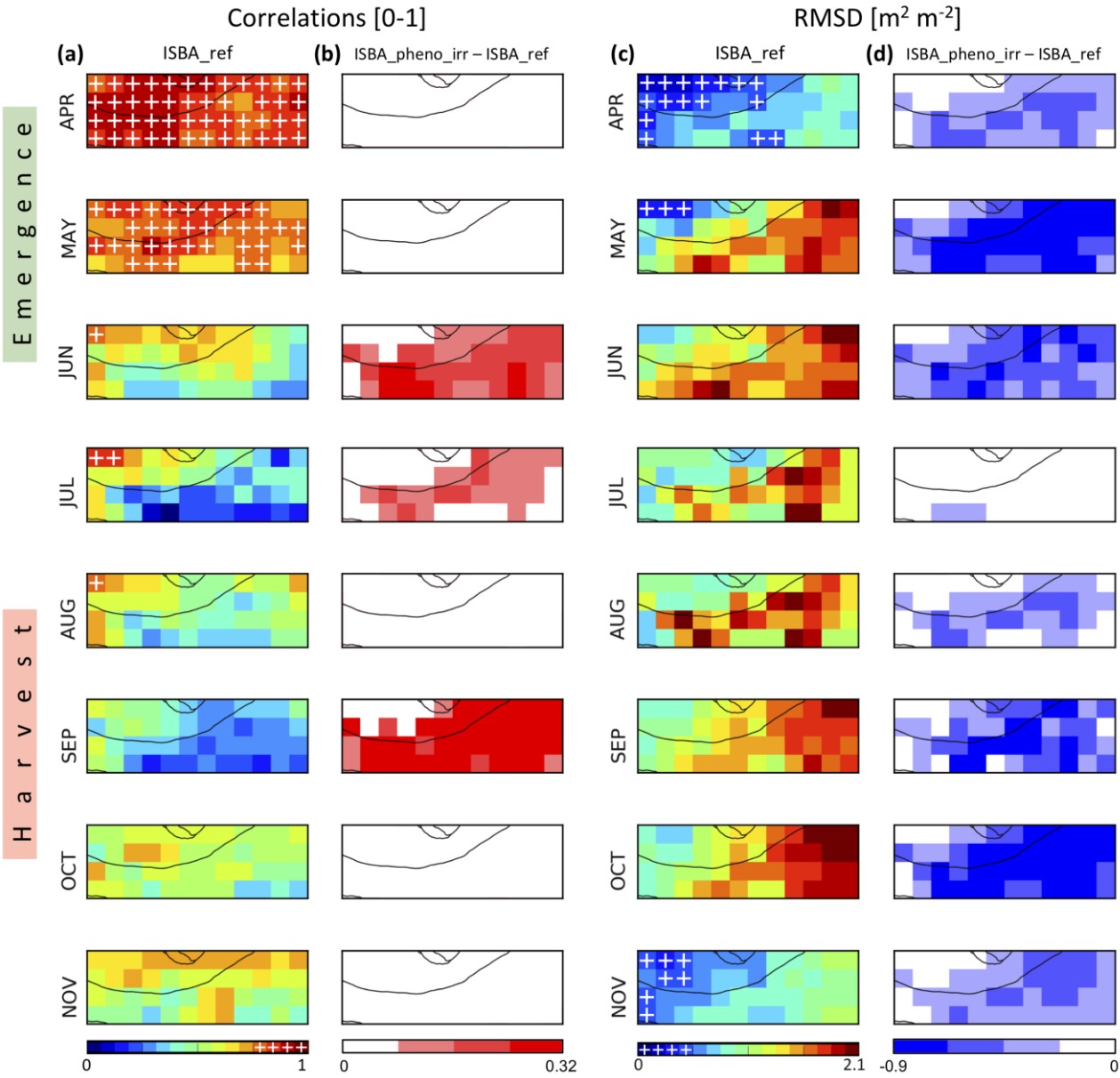

**Figure 7** – **Comparison of simulated LAI with CGLS LAI observations in the most densely irrigated part of Nebraska (Fig. 1e) from 1999 to 2018 during the vegetation growing and senescence time period from April to November. Monthly temporal correlation (a, b) and RMSD (c, d) maps are shown for the reference simulation without a representation of irrigation ISBA_ref (a, c). The added value of the ISBA_pheno_irr simulation with respect to ISBA-ref is shown through score difference maps (b, d). ISBA_ref correlations (RMSD) values larger (smaller) than 0.75 (0.525 $m^2m^{-2}$) are indicated by white plus symbols.**

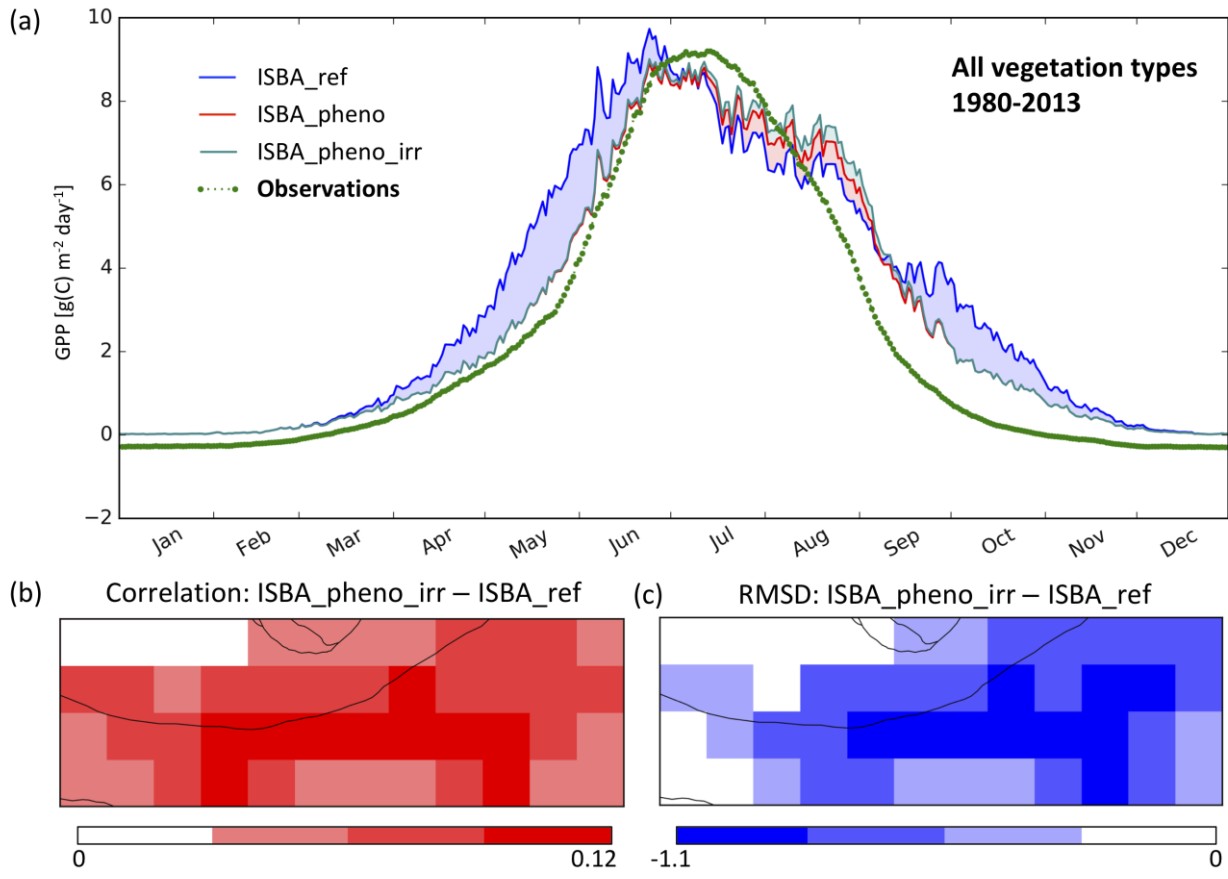


**Figure 8 – Seasonal variation of mean daily GPP values (gC m⁻² d⁻¹) in the most densely irrigated part of Nebraska (Fig. 1e) from 1980 to 2013 (a) as derived from the reference simulation ISBA_ref (blue line), ISBA_pheno (red line), ISBA_pheno_irr (cyan) and observations (green dotted line). Temporal correlation (b) and RMSD (c) score difference maps show the added value**

**of the ISBA_pheno_irr simulation with respect to ISBA-ref.**

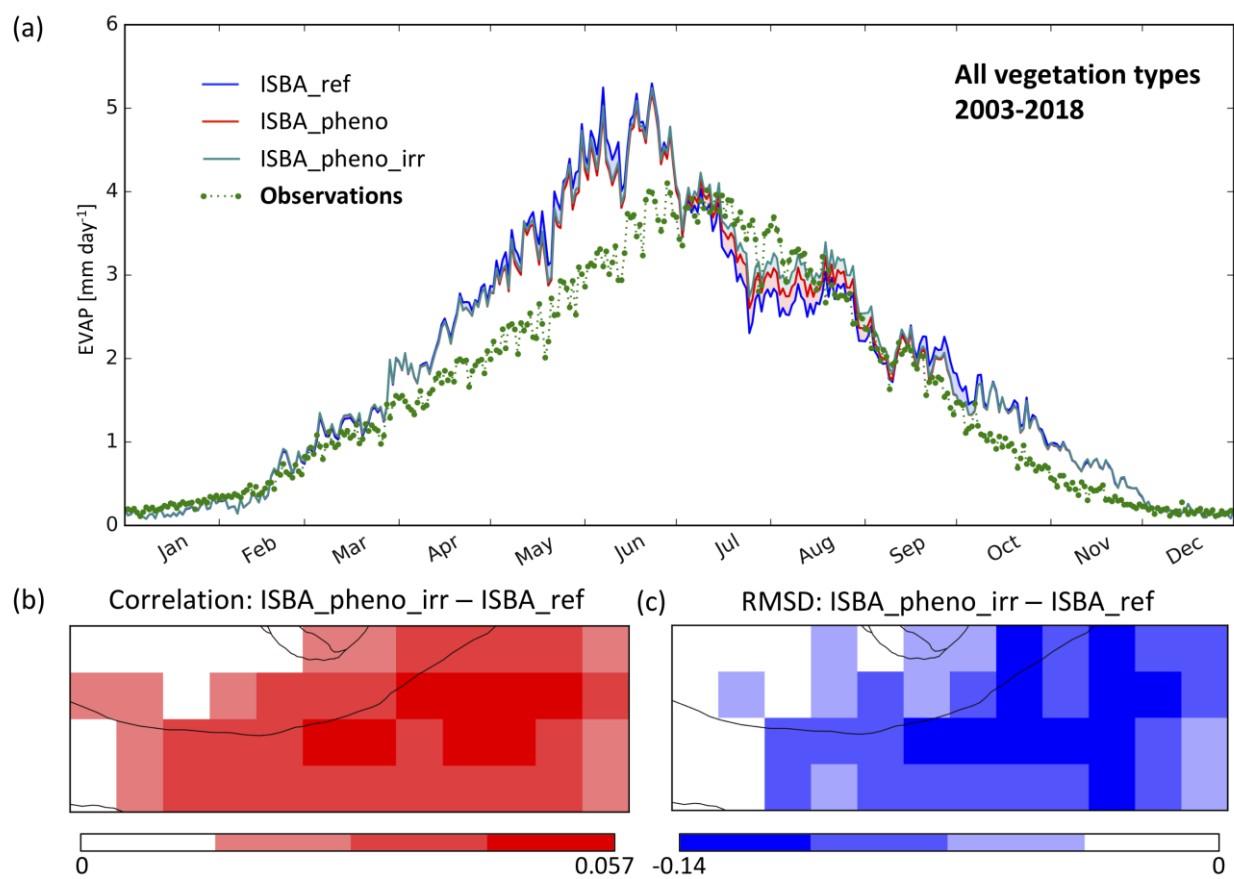

**Figure 9** – **Seasonal variation of mean daily evapotranspiration values (kg m⁻² d⁻¹) in the most densely irrigated part of Nebraska (Fig. 1e) from 2003 to 2018 (a) as derived from the reference simulation ISBA_ref (blue line), ISBA_pheno (red line), ISBA_pheno_irr (cyan) and GLEAM observations (green dotted line). Temporal correlation (b) and RMSD (c) score difference maps show the added value of the ISBA_pheno_irr simulation with respect to ISBA-ref.**

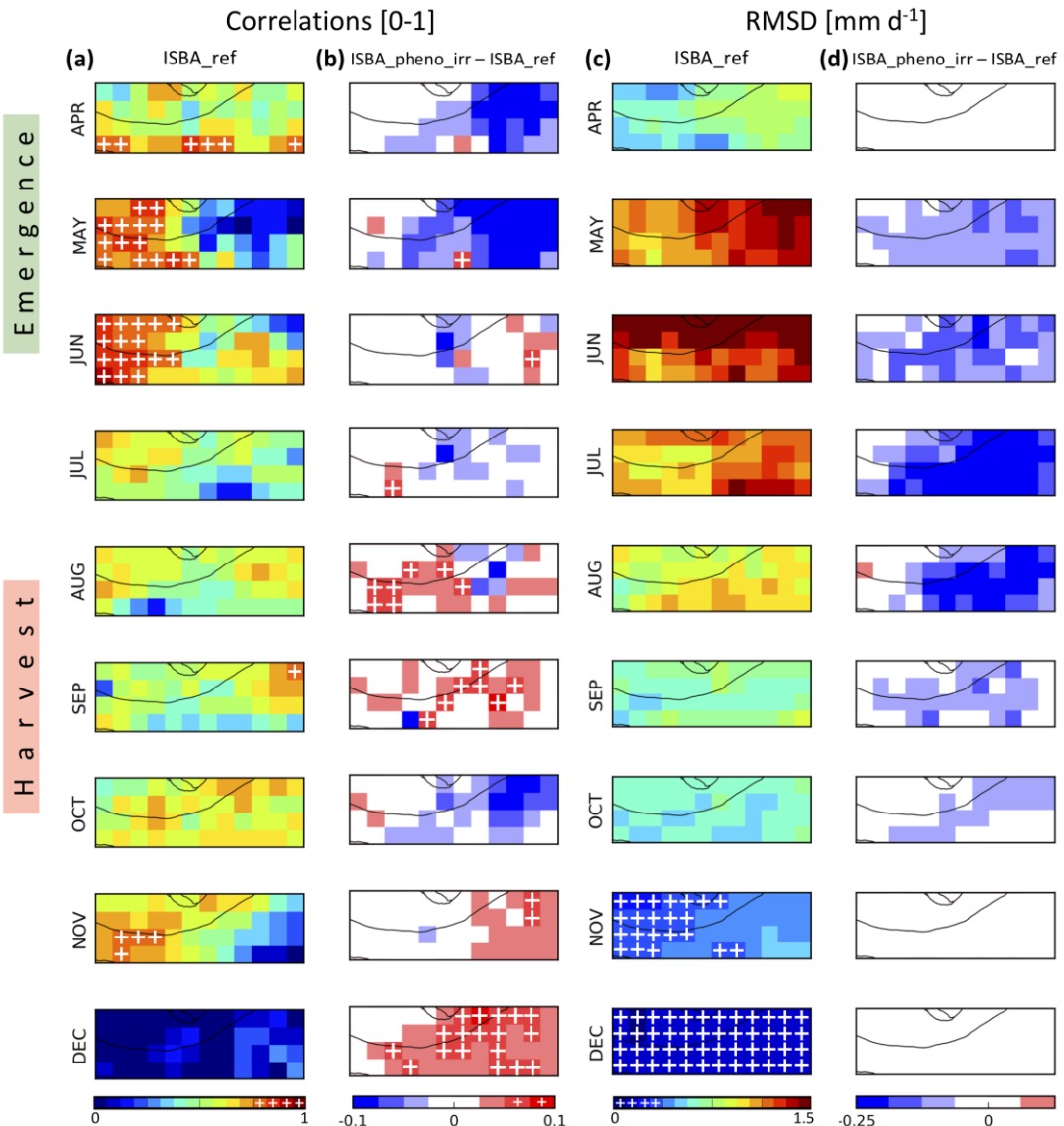

**Figure 10 – Comparison of simulated evapotranspiration with GLEAM evapotranspiration observations in the most densely irrigated part of Nebraska (Fig. 1e) from 2003 to 2018 during the vegetation growing and senescence time period from April to November. Monthly temporal correlation (a, b) and RMSD (c, d) maps are shown for the reference simulation without a representation of irrigation ISBA_ref (a, c). The added value of the ISBA_pheno_irr simulation with respect to ISBA-ref is shown through score difference maps (b, d). ISBA_ref correlations (RMSD) values larger (smaller) than 0.75 (0.375 mm d⁻¹) are indicated by white plus symbols. Correlations differences larger than 0.05 are indicated by white plus symbols.**

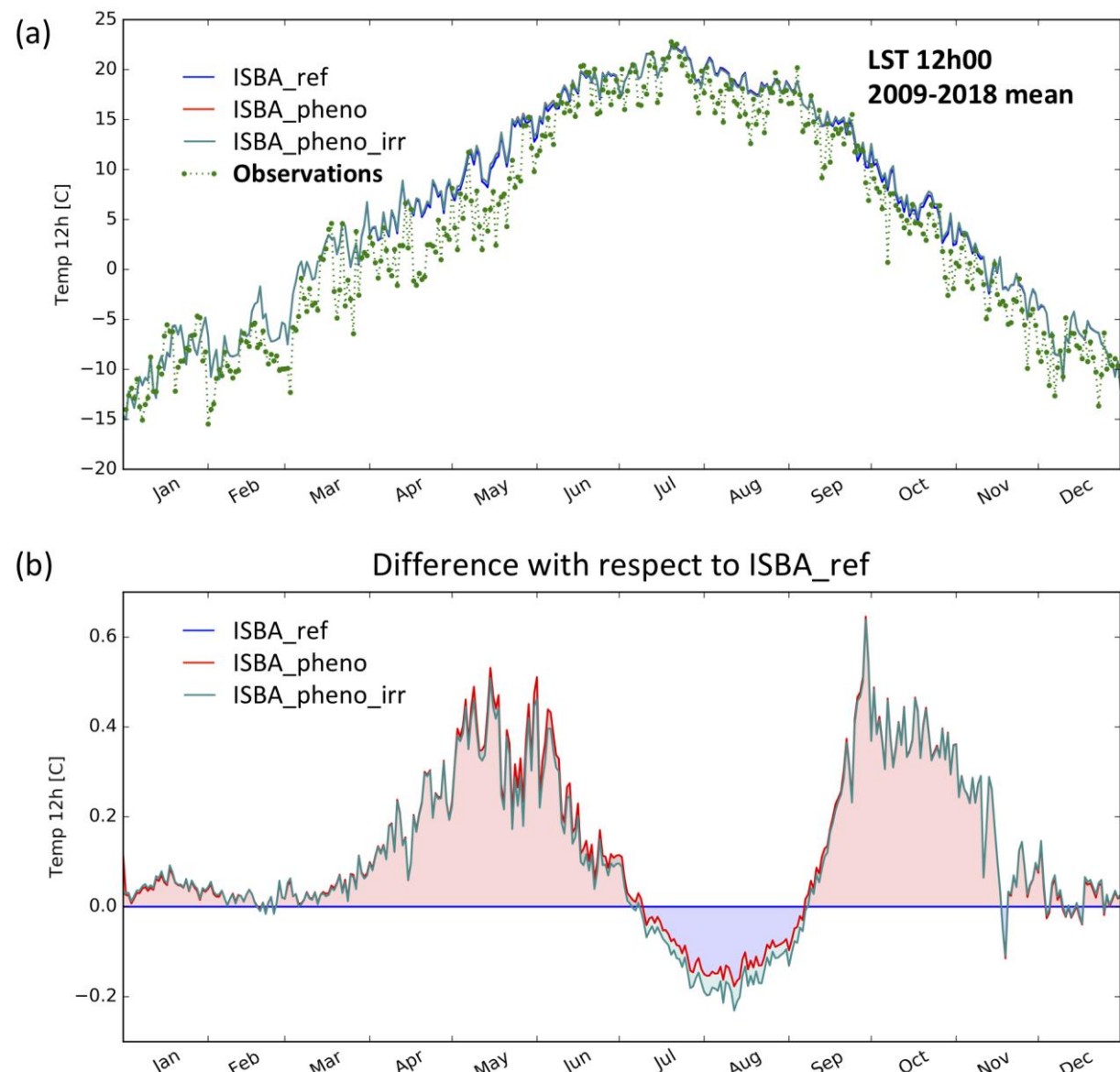

**Figure 11 – Seasonal variation of surface temperature daily values at 12:00 local time (degree C) in the most densely irrigated part of Nebraska (Fig. 1e) from 2009 to 2018 (a) as derived from the reference simulation ISBA_ref (blue line), ISBA_pheno (red line), ISBA_pheno_irr (cyan) and the CGLS product (green dotted line). The surface temperature differences at 12:00 local time (b) of ISBA_pheno_irr and ISBA_pheno simulations with respect to the ISBA-ref simulations are shown.**

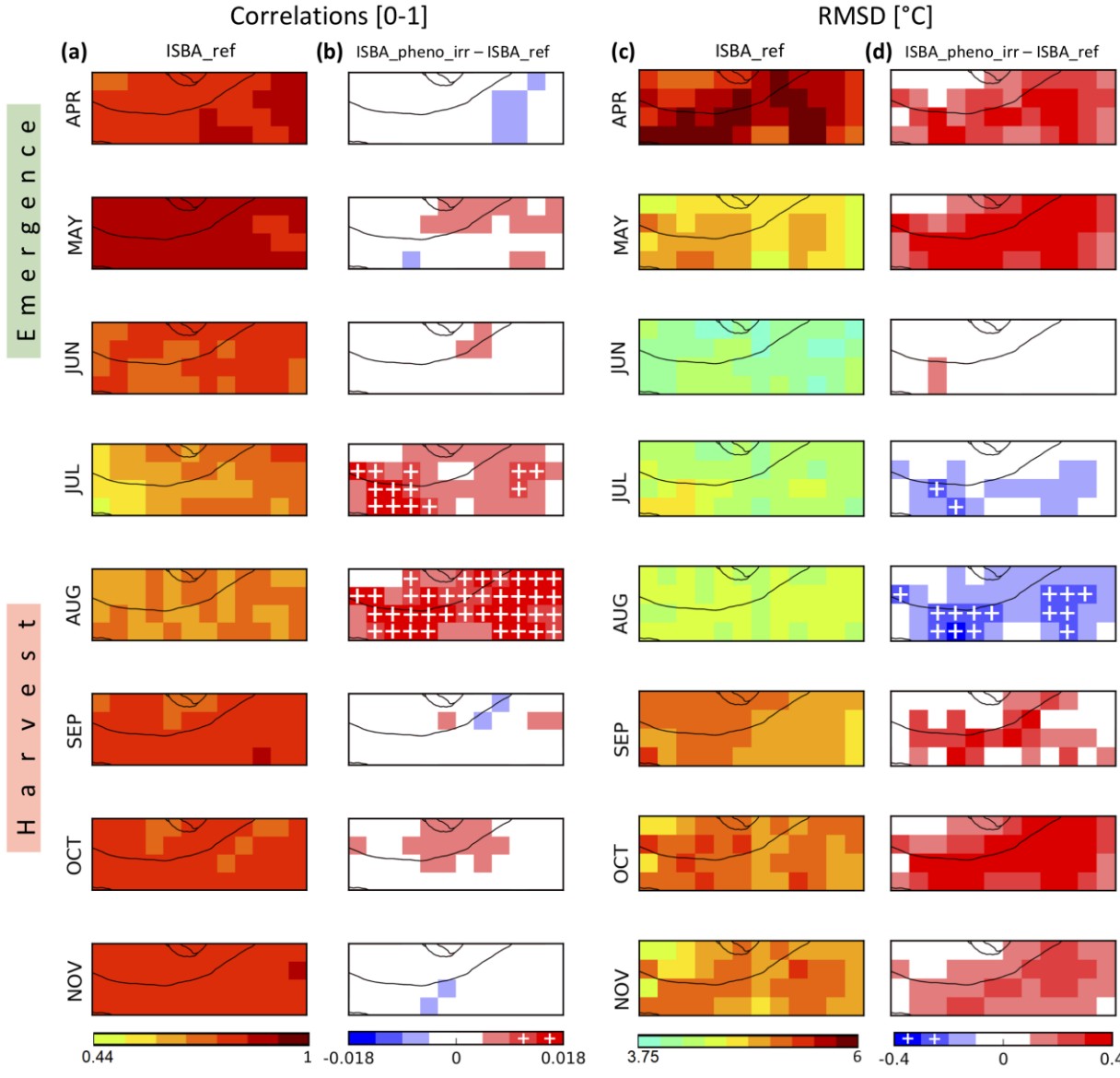

**Figure 12 – Comparison of simulated surface temperature daily values at 12:00 local time with CGLS observations in the most densely irrigated part of Nebraska (Fig. 1e) from 2009 to 2018 during the vegetation growing and senescence time period from April to November. Monthly temporal correlation (a, b) and RMSD (c, d) maps are shown for the reference simulation without a representation of irrigation ISBA_ref (a, c). The added value of the ISBA_pheno_irr simulation**

with respect to ISBA-ref is shown through score difference maps (b, d). Correlations (RMSD) differences larger (smaller) than 0.009 (-0.2°C) are indicated by white plus symbols.