# Peer review of "Implementation of a new crop and irrigation scheme in the ISBA land surface model using SURFEX\_v8.1"

_Geoscientific Model Development, 2021_

## Referee Comment (RC1)

General comments:

The authors incorporated an irrigation scheme into the ISBA LSM and compared it with satellite-based data to evaluate the improvement in accuracy. As the authors pointed out, the consideration of an irrigation scheme in LSMs is one of important challenges, and it will be a great contribution to this field if it is properly demonstrated. However, it is difficult to say that the authors' objective has been achieved in the manuscript in the following points. Therefore, a major revision or rejection seems appropriate.

(1): The results presented indicate the improvement in accuracy due to the introduction of crop phenology such as emergence and harvest, rather than that of an irrigation scheme

(2): Even with the introduction of crop phenology and an irrigation scheme, there is still a large difference from observations, especially in seasonal changes. The major reason for this difference may be that appropriate validation data are not used rather than model problems. For example, the observational LAI change shown in Figure 4 is odd for a crop LAI change (the model LAI change is more plausible). To solve this problem, it would be appropriate to compare the model output with the site-scale LAI and GPP observed on the farm.

Minor comments:

L66–69: Irrigation schemes have been integrated into several large scale LSMs such as CLM. More intensive literature review on the topic is needed.

L90: Brief description on the SURFEX is needed here.

L139–140: Describe lon. and lat. of the two places in the same way.

L158: Is simulated LST soil surface temperature under the canopy? Can satellite measure it?

L195: What is difference between the irrigation amount and the irrigation rate?

L216: through?

Section 2.4: It is better to place this section at the beginning of Section 2 for easier understanding.

L314: Clarify ISBA_ref does not include crop phenology

Section 2: The detail of crop phenology and LAI development should be described.

Section 3.1: Simply compare irrigation water amount between observations and simulations, and show the correlation and significance.

---

## Referee Comment (RC2)

General comments:

The authors developed a new irrigation scheme in the ISBA land surface model, in which several parameters could be assigned by input. This is of great importance, as in simulations, the spatial heterogeneity of the irrigation application could be taken into account with this new scheme. However, not all properties of this new scheme have been presented, like the irrigation methods, irrigation rate and irrigation time. It seems that the authors made a lot of efforts to prove that the new irrigation scheme is better than the old one, while this is always challenging, as there are always some other factors that may affect the results. I am not surprised that the results are not convincing enough. As we can see, in this study the seasonal patterns of LAI and GPP do not match the observations, and a possible reason may be that the crop growth module of the model is not suitable for the crops in this region, and there are few that the new irrigation scheme could do with this problem. Thus, it is very important to know what scientific question the authors are addressing here: a new irrigation scheme which considers spatial heterogeneity of irrigation, or a new irrigation scheme which is more suitable for the region of Nebraska? For me, the authors should show more this new irrigation scheme could do, and how it will affect the outputs of the model, rather than insisting on the superiority of the new scheme. Finally, the authors need to restructure some parts and rewrite some sentences. To be accepted, a major revision would be necessary for this study.

Minor comments:

L92: add the reasons why you chose to only consider offline simulations.

L95: this part is the description of study area, so I believe it can be moved to Section 2.

L105: It's a bit weird wo have this part here, I would move it after the description of model and the new irrigation scheme.

L115: specify a bit more how this rule is applied here.

L131: give the reasons why you chose this period.

L195: be simulated or be assigned?

L238: what is the difference between drip and flood irrigation?

L321: since the USGS provide data every five year, then it is possible to compare the yearly irrigation water amount rather than the multi-year averaged value.

L327: It is always challenging to evaluate a model by comparing the model output to satellite-based fluxes data, as it is not easy to validate the quality of the data. My suggestion would be doing a single point run, and comparing the results with the station-observed data.

Figure1: Add the lat-lon grid on the frames.

Figure5: Specify that positive value for Correlation (b) means that the result of ISBA_pheno_irr is better, and for RMSD(c) negative value means that result of ISBA_pheno_irr is better. Otherwise it could be a bit confused.

---

## Author Comment (AC2)

**Reviewer #1**

**Comment R1.1:**

*The results presented indicate the improvement in accuracy due to the introduction of crop phenology such as emergence and harvest, rather than that of an irrigation scheme.*

**Response R1.1:**

**We agree. The introduction of crop phenology processes such as emergence and harvest has a marked impact on the results, even without activating irrigation.**

**We propose to change the title of the paper from**

**"Implementation and validation of a new irrigation scheme in the ISBA land surface model"**

**to**

**"Implementation and validation of a new crop and irrigation scheme in the ISBA land surface model".**

**Throughout the manuscript,**

**"new irrigation …"**

**was replaced by**

**"new crop and irrigation …".**

**Comment R1.2:**

*Even with the introduction of crop phenology and an irrigation scheme, there is still a large difference from observations, especially in seasonal changes. The major reason for this difference may be that appropriate validation data are not used rather than model problems. For example, the observational LAI change shown in Figure 4 is odd for a crop LAI change (the model LAI change is more plausible). To solve this problem, it would be appropriate to compare the model output with the site-scale LAI and GPP observed on the farm.*

**Response R1.2:**

**We agree.**

**Corn is the dominant crop type in the considered irrigated area in Nebraska (Zhang et al. 2020). While the satellite LAI observations present a peak at the end of July, the modelled LAI is plateauing in August (Fig. 4). Corn LAI observations at the field scale for various agricultural management conditions are showed in Boedhram et al. (2001). These data show that the modelled LAI plateau in August at LAI values of about 3.5 m2m-2 is realistic.**

In Table 4, we included Boedhram et al. (2001) LAI data for fertilized irrigated corn in 1994 and 1995.

"Corn is the dominant crop type in the considered irrigated area in Nebraska (Zhang et al. 2020). While the satellite LAI observations present a peak at the end of July, the modelled LAI is plateauing in August (Fig. 4). Corn LAI observations at the field scale for various agricultural management conditions are showed in Boedhram et al. (2001). These data show that the modelled LAI plateau in August at LAI values of about 3.5 m2m-2 is realistic."

will be added to Section 3.1.

Table 4 – Simulated mean LAI peak characteristics over Nebraska for the 1999-2018 time period for crops (see Fig. 4) and all vegetation types (see Fig. 5), together with satellite and in situ observations.

| Vegetation types | LAI source | Peak LAI ($m^2 m^{-2}$) | Peak LAI date |
|---|---|---|---|
| Crops | Satellite observations | 4.9 (±0.8) | 31 July |
| | Boedhram et al. 2001 (*) | 3.6 to 4.0 | 12 July to 19 August 1994 |
| | Boedhram et al. 2001 (*) | 3.5 | 2 August to 23 August 1995 |
| | ISBA_ref | 3.6 (±0.2) | 2 July |
| | ISBA_pheno | 3.5 (±0.2) | 26 August |
| | ISBA_pheno_irr | 3.7 (±0.1) | 28 August |
| All | Satellite observations | 3.8 (±1.5) | 31 July |
| | ISBA_ref | 3.3 (±0.3) | 1 July |
| | ISBA_pheno | 3.1 (±0.3) | 16 July |
| | ISBA_pheno_irr | 3.1 (±0.3) | 16 July |

(*)Boedhram et al. (2001) data are for fertilized irrigated corn in 1994 and 1995.

**References:**

Boedhram, N., T. J. Arkebauer, and W. D. Batchelor: Season-long characterization of vertical distribution of leaf area in corn, Agron. J., 93, 1235–1242, https://doi.org/10.2134/agronj2001.1235, 2001.

Zhang, Z., M. Barlage, F. Chen, Y. Li, W. Helgason, X. Xu, X. Liu, and Z. Li: Joint modeling of crop and irrigation in the central United States using the Noah-MP land surface model. Journal of Advances in Modeling Earth Systems, 12, e2020MS002159, https://doi.org/10.1029/2020MS002159, 2020.

**Comment R1.3:**

*L66–69: Irrigation schemes have been integrated into several large scale LSMs such as CLM. More intensive literature review on the topic is needed.*

**Response R1.3:**

**Yes. We will introduce recent references in the Introduction describing the implementation over the USA of a representation of crop and irrigation into land surface models: Zhang et al. for Noah-MP, and Felfelani et al. (2020) for CLM.**

**References:**

**Felfelani, F., D. M. Lawrence, and Y. Pokhrel: Representing intercell lateral groundwater flow and aquifer pumping in the community land model, Water Resources Research, 56, e2020WR027531, https://doi.org/10.1029/2020WR027531, 2020.**

**Zhang, Z., M. Barlage, F. Chen, Y. Li, W. Helgason, X. Xu, X. Liu, and Z. Li: Joint modeling of crop and irrigation in the central United States using the Noah-MP land surface model. Journal of Advances in Modeling Earth Systems, 12, e2020MS002159, https://doi.org/10.1029/2020MS002159, 2020.**

**Comment R1.4:**

*L90: Brief description on the SURFEX is needed here.*

**Response R1.4:**

**The following sentence was added:**

**"SURFEX integrates different models describing ocean and terrestrial surfaces. Over land, specific models are used to represent water bodies, cities, and the soil-plant system. The latter is modelled by the ISBA LSM."**

**Comment R1.5:**

*L139–140: Describe lon. and lat. of the two places in the same way.*

**Response R1.5:**

**"40.83°N, 96.76°W"**

**was replaced by**

**"40.83°N - 96.76°W" .**

**Comment R1.6:**

*L158: Is simulated LST soil surface temperature under the canopy? Can satellite measure it?*

**Response R1.6:**

**In the version of the model used in this study, a single composite soil-vegetation energy budget is used and the thermal effect of crop residues is not represented. This means that over vegetated areas, the simulated LST can differ from the vegetation temperature as seen from space.**

**This will be added to the manuscript.**

**Comment R1.7:**

*L195: What is difference between the irrigation amount and the irrigation rate?*

**Response R1.7:**

**"A number of irrigation variables need to be simulated such as the irrigation amount, the irrigation rate"**
**was replaced by**

**"A number of irrigation parameters need to be assigned such as the irrigation amount, the irrigation interval".**

**Comment R1.8:**

*L216: through?*

**Response R1.8:**

**Yes, thanks for noting this.**

**Comment R1.9:**

*Section 2.4: It is better to place this section at the beginning of Section 2 for easier Understanding.*

**Response R1.9:**

**We agree. Section 2.4 will be moved to the beginning of Section 2 together with Sections 2.2 and 2.3. Section 2.1 will be placed at the end of Section 2.**

**Comment R1.10:**

*L314: Clarify ISBA_ref does not include crop phenology.*

**Response R1.10:**

**""ISBA_ref" without any irrigation (the benchmark)"**

**was replaced by**

**""ISBA_ref" without irrigation nor crop phenology (the benchmark)"**

**Comment R1.11:**

*Section 2: The detail of crop phenology and LAI development should be described.*

**Response R1.11:**

**A specific subsection on crop phenology will be added. It will be indicated that in ISBA_ref, phenology is entirely driven by photosynthesis and that no growing degree-day model is used. The only phenology parameter is a minimum LAI value of 0.3 $m^2m^{-2}$ for low vegetation. In this study, two more parameters are used: emergence and harvest dates (Table 2). After the harvest and before the emergence, the simulated LAI is maintained at the minimum LAI value of 0.3 $m^2m^{-2}$.**

**Comment R1.12:**

*Section 3.1: Simply compare irrigation water amount between observations and simulations, and show the correlation and significance.*

**Response R1.12:**

**A direct comparison would not have been statistical significant because complete USGS observations were available only for 6 years during the considered time period.**

---

## Author Comment (AC3)

**Reviewer #2**

**Comment R2.1:**

*The authors developed a new irrigation scheme in the ISBA land surface model, in which several parameters could be assigned by input. This is of great importance, as in simulations, the spatial heterogeneity of the irrigation application could be taken into account with this new scheme. However, not all properties of this new scheme have been presented, like the irrigation methods, irrigation rate and irrigation time. It seems that the authors made a lot of efforts to prove that the new irrigation scheme is better than the old one, while this is always challenging, as there are always some other factors that may affect the results. I am not surprised that the results are not convincing enough. As we can see, in this study the seasonal patterns of LAI and GPP do not match the observations, and a possible reason may be that the crop growth module of the model is not suitable for the crops in this region, and there are few that the new irrigation scheme could do with this problem. Thus, it is very important to know what scientific question the authors are addressing here: a new irrigation scheme which considers spatial heterogeneity of irrigation, or a new irrigation scheme which is more suitable for the region of Nebraska? For me, the authors should show more this new irrigation scheme could do, and how it will affect the outputs of the model, rather than insisting on the superiority of the new scheme. Finally, the authors need to restructure some parts and rewrite some sentences.*

**Response R2.1:**

**Many thanks for your comments. I the revised version of the manuscript, we strived to clarify the objectives of the paper and to improve its structure (see also Responses to the Editor and to Reviewer 1).**

**Comment R2.2:**

*L92: add the reasons why you chose to only consider offline simulations.*

**Response R2.2:**

**"While the SURFEX framework allows the coupling of terrestrial processes with atmospheric and hydrological models, only offline ISBA simulations are considered in this study. The evaluation of the new irrigation scheme is made over the state of Nebraska (United States of America, USA). This area presents a high density of irrigated fields (Fig. 1) and large freely available observational datasets for evaluation."**

**was replaced by**

**"In the SURFEX platform, the ISBA model can be coupled to the CTRIP model (Decharme et al., 2019, Munier and Decharme, 2021) which is specifically designed to represent water dynamics within rivers and aquifers. The SURFEX framework allows the coupling of terrestrial processes with atmospheric models and hydrological models. For agricultural drought and water resource monitoring, SURFEX can also be operated of-**

fline, forced by a pre-existing dataset of atmospheric variables. Only offline ISBA simulations are considered in this study. The new irrigation module represents water demand for irrigation, only, and irrigation is not limited by the lack of water resources. This has consequences on water conservation. However, water used for irrigation is usually withdrawn from aquifers, rivers or reservoirs. These compartments are not re-presented in ISBA but a new module dedicated to dam/reservoirs is currently under development. The evaluation of the new irrigation scheme is made over the state of Nebraska (United States of America, USA). This area presents a high density of irrigated fields (Fig. 1) and large freely available observational datasets for evaluation."

**References:**

Decharme, B., Delire, C., Minvielle, M., Colin, J., Vergnes, J. P., Alias, A., ... & Voldoire, A. (2019). Recent changes in the ISBA-CTRIP land surface system for use in the CNRM-CM6 climate model and in global off-line hydrological applications. Journal of Advances in Modeling Earth Systems, 11(5), 1207-1252.

Munier, S., & Decharme, B. (2021). River network and hydro-geomorphology parametrization for global river routing modelling at 1/12° resolution. Earth System Science Data Discussions, 1-28.

**Comment R2.3:**

*L95: this part is the description of study area, so I believe it can be moved to Section 2.*

**Response R2.3:**

**"The evaluation of the new irrigation scheme is made over the state of Nebraska (United States of America, USA). This area presents a high density of irrigated fields (Fig. 1) and large freely available observational datasets for evaluation."**

**Was moved to Section 2.**

**Comment R2.4:**

*L105: It's a bit weird wo have this part here, I would move it after the description of model and the new irrigation scheme.*

**Response R2.4:**

**We agree. Section 2.1 will be placed at the end of Section 2. Section 2.4 will be moved to the beginning of Section 2 together with Sections 2.2 and 2.3.**

**Comment R2.5:**

*L115: specify a bit more how this rule is applied here.*

**Response R2.5:**

**Yes. "spatial rescaling" was replaced by "spatial resampling".**

The 300 m × 300 m resampled irrigation map was published on zenodo (https://doi.org/10.5281/zenodo.6011618).

**Comment R2.6:**

*L131: give the reasons why you chose this period.*

**Response R2.6:**

**"A subset of the ERA-5 forcing over Nebraska was used for the time period from 1979 to 2018."**
**was replaced by**

**"A subset of the ERA-5 forcing over Nebraska was used for the time period from 1979 to 2018. This period was chosen in order to encompass various validation datasets."**

**Comment R2.7:**

*L195: be simulated or be assigned?*

**Response R2.7:**

**"A number of irrigation variables need to be simulated such as the irrigation amount, the irrigation rate"**
**was replaced by**

**"A number of irrigation parameters need to be assigned such as the irrigation amount, the irrigation interval".**

**Comment R2.8:**

*L238: what is the difference between drip and flood irrigation?*

**Response R2.8:**

**"In this study, only sprinkling irrigation is considered."**

**was replaced by**

**"In this study, only sprinkling irrigation is considered as this is the dominant irrigation type in Nebraska. Drip and flood irrigation will be evaluated in future works. The activation of a given irrigation method is described in Supplement 5. For sprinkling irrigation, water is added to the precipitation forcing. For drip and flood irrigation, the water flux is applied directly to the soil surface with no leaf interception as explained in section 2.3.1. Considering the static equipment used for drip irrigation, there is no irrigation interval ($\Delta t_{Wn}$ = 0 day)."**

**Comment R2.9:**

*L321: since the USGS provide data every five year, then it is possible to compare the yearly irrigation water amount rather than the multi-year averaged value.*

**Response R2.9:**

**A direct comparison would not have been statistical significant because complete USGS observations were available only for 6 years during the considered time period.**

**Comment R2.10:**

*L327: It is always challenging to evaluate a model by comparing the model output to satellite-based fluxes data, as it is not easy to validate the quality of the data. My suggestion would be doing a single point run, and comparing the results with the station-observed data.*

**Response R2.10:**

**We included new elements in Table 4 showing the added-value of crop and irrigation options on LAI simulation. See response R1.2 to Reviewer 1.**

**Comment R2.11:**

*Figure1: Add the lat-lon grid on the frames.*

**Response R2.11:**

**This is a new version of Figure 1, incorporating lat-lon grids:**

[Figure]

**Comment R2.12:**

*Figure5: Specify that positive value for Correlation (b) means that the result of ISBA_pheno_irr is better, and for RMSD (c) negative value means that result of ISBA_pheno_irr is better. Otherwise it could be a bit confused.*

**Response R2.12:**

**A sentence was added to Section 2 specifying this.**

---

## Author Comment (AC4)

**Editor**

*This study describes a new irrigation parameterization for ISBA land surface model (LSM). Despite interesting subjective of this study, it is hard to read and to understand what the unique things in this study are. Please revise the manuscript carefully for better readability and provide clearly 1) how to parameterize irrigation processes in the model codes, 2) remove redundant sentences many places, 3) clarify the information on the data so that other people reproduce what this study did, and 4) rewrite fractured sentences. It is also important to organize sentences, paragraph, and figures to converge into clear goals and to support your conclusions. Here I put some comments on the manuscript, but I believe that overall structure of this manuscript should be reorganized and rewritten carefully.*

**Response:**

**Many thanks for your in-depth review of the manuscript and for your comments. We strived to highlight the novel aspects of this work and to improve the organization of the paper (see below and see responses to reviewers 1 and 2).**

**Comment E.1:**

*There is not enough information to reproduce the modeling results in this manuscript. Please provide more details on the irrigation parameterization especially for different kinds of irrigation methods. How do you deal with different irrigation types in the model?*

**Response E.1:**

**In this study, only sprinkling irrigation is considered. All the irrigation parameters needed to launch the simulation are listed in Table 2. The activation of a given irrigation method is described in Supplement 5, with two examples showing how to launch a simulation. In order to improve the reproducibility of our results we have included a doi reference pointing to the SURFEX initialization files and to the spatially resampled irrigation map (https://doi.org/10.5281/zenodo.6011618).**

**For sprinkling irrigation, water is added to the precipitation forcing. For drip and flood irrigation, the water flux is applied directly to the soil surface with no leaf interception as explained in section 2.3.1. Considering the static equipment used for drip irrigation, there is no irrigation interval ($\Delta t_{Wn}$ = 0 day). This was indicated in the revised version of the manuscript, Section 2, and in Table 2:**

**"In this study, only sprinkling irrigation is considered."**

**was replaced by**

**"In this study, only sprinkling irrigation is considered as this is the dominant irrigation type in Nebraska. Drip and flood irrigation will be evaluated in future works. The activation of a given irrigation method is described in Supplement 5. For sprinkling irrigation, water is added to the precipitation forcing. For drip and flood irrigation, the water flux is applied directly to the soil surface with no leaf interception as explained in**

**section 2.3.1. Considering the static equipment used for drip irrigation, there is no irrigation interval ($\Delta t_{Wn}$ = 0 day).”**

**For the sake of clarity, the following decision tree Figure, valid for all irrigation types, was added to the manuscript:**

[Figure]

**For the sake of clarity, the following Figures showing the fraction of irrigated C3 and C4 crops were added to the Supplement:**

[Figure]

**Comment E.2:**

*Please provide specific information on water conservation and differences between different irrigation methods in the model.*

**Response E.2:**

**The present irrigation module in ISBA represents water demand for irrigation, only, and irrigation is not limited by the lack of water resources. This indeed has consequences on water conservation. However, water used for irrigation is usually withdrawn from aquifers, rivers or reservoirs. These compartments are not represented in ISBA. In the SURFEX platform, the ISBA model can be coupled to the CTRIP model (Decharme et al., 2019, Munier and Decharme, 2021) which is specifically designed to represent water dynamics within rivers and aquifers. In addition, a new module dedicated to dam/reservoirs is currently under development. Future work will focus on the coupling between the new irrigation module in ISBA and CTRIP, thus ensuring the water conservation.**

**In section 1,**

**"While the SURFEX framework allows the coupling of terrestrial processes with atmospheric and hydrological models, only offline ISBA simulations are considered in this study. The evaluation of the new irrigation scheme is made over the state of Nebraska (United States of America, USA). This area presents a high density of irrigated fields (Fig. 1) and large freely available observational datasets for evaluation."**

**was replaced by**

**"In the SURFEX platform, the ISBA model can be coupled to the CTRIP model (Decharme et al., 2019, Munier and Decharme, 2021) which is specifically designed to represent water dynamics within rivers and aquifers. The SURFEX framework allows the coupling of terrestrial processes with atmospheric models and hydrological models. For agricultural drought and water resource monitoring, SURFEX can also be operated offline, forced by a pre-existing dataset of atmospheric variables. Only offline ISBA simulations are considered in this study. The new irrigation module represents water demand for irrigation, only, and irrigation is not limited by the lack of water resources. This has consequences on water conservation. However, water used for irrigation is usually withdrawn from aquifers, rivers or reservoirs. These compartments are not re-presented in ISBA but a new module dedicated to dam/reservoirs is currently under development. The evaluation of the new irrigation scheme is made over the state of Nebraska (United States of America, USA). This area presents a high density of irrigated fields (Fig. 1) and large freely available observational datasets for evaluation."**

**References:**

**Decharme, B., Delire, C., Minvielle, M., Colin, J., Vergnes, J. P., Alias, A., ... & Voldoire, A. (2019). Recent changes in the ISBA-CTRIP land surface system for use in the CNRM-CM6 climate model and in global off-line hydrological applications. Journal of Advances in Modeling Earth Systems, 11(5), 1207-1252.**

Munier, S., & Decharme, B. (2021). River network and hydro-geomorphology parametrization for global river routing modelling at 1/12° resolution. Earth System Science Data Discussions, 1-28.

**Comment E.3:**

*Writing is important and this manuscript is not well organized. This manuscript is not easy to read and understand because of inconsistent and poorly organized sentences and redundant statements. More efforts are necessary to revise the manuscript carefully.*

*- Line 176: One example of redundant sentences in this manuscript (« Moreover, an updated land cover description was used: ECOCLIMAP-SG (see Supplement S1) »).*

*- Line 188: One example of redundant sentences in this manuscript (« The new irrigation scheme is operated using the ECOCLIMAP-SG land cover classification within SURFEX »).*

*- Line 254: One example of redundancy*

*- Line 304: One example of redundancy (« The ISBA LSM simulations (non-coupled with the atmosphere) are forced by the ERA-5 reanalysis »).*

**Response E.3:**

**We agree. We did our best to account for your remarks as well as remarks from Reviewers 1 and 2. You can find below the response to your specific comments.**

**On L. 176, « Moreover, an updated land cover description was used: ECOCLIMAP-SG (see Supplement S1) » was deleted.**

**On L. 188, « The new irrigation scheme is operated using the ECOCLIMAP-SG land cover classification within SURFEX » was replaced by « The new irrigation scheme is operated using the ECOCLIMAP-SG land cover classification within SURFEX (see Supplement S1). »**

**On L. 304, « non-coupled with the atmosphere » was deleted.**

**We were not able to find a redundancy on L. 254.**

**Comment E.4:**

*Fluxcom data by Jung et al. do not consider the irrigation process and I am not sure if these data are useful for the model evaluation.*

**Response E.4:**

**This is a very good point. We tend to believe that the FLUXCOM data are relevant over irrigated areas at low spatial resolution. Al-Yaari et al. 2021 showed that the FLUXCOM daily evapotranspiration product can be used as a benchmark over irrigated areas. They compared global evapotranspiration simulations of the ORCHIDEE land surface model with FLUXCOM without activating an irrigation module in ORCHIDEE. They found**

that a negative model bias can be observed over irrigated areas while the model is virtually unbiased over rainfed areas. The negative bias increases as the irrigation fraction increases, suggesting that FLUXCOM is sensitive to irrigation. The information on irrigation could come from the remote sensing data incorporated into the FLUXCOM products. Since evapotranspiration and GPP fluxes are closely connected to each other, it can be assumed that the FLUXCOM GPP product is also sensitive to irrigation.

This will be indicated in the revised version of the manuscript.

[Figure]

**Figure 3 – Yearly cumulated number of irrigation events simulated by the model for the studied area in Nebraska from 1979 to 2018 (blue dots). The six yearly estimates from USGS for 1985, 1990, 2000, 2005, 2010, and 2015 are indicated (red triangles). The mean and standard deviation of the yearly values are shown for the model (green solid and dashed lines, respectively), and for the USGS water data from 1985 to 2015 (brown lines).**

**Comment E.9:**

*Line147: What kinds of inconsistencies? Any impacts on the results and conclusion?*

**Response E.9:**

**For 1995 conveyance loss data are not available.**

**"The USGS data we used cover the 1985-2019 time period. Because of inconsistencies in the record for 1995, this year was not taken into account."**

**was replaced by**

**"The USGS data we used cover the 1985-2015 time period. Because conveyance loss data are missing in the record for 1995, this year was not taken into account."**

**Comment E.10:**

*Line 149: What LAI values are used for the initial conditions?*

**Response E.10:**

**As explained on L. 305, a spin-up simulation was made. The same initial conditions are used for all the simulations, with and without crop and irrigation modeling. Section 2 was reorganized and model implementation is now described before describing validation datasets. Initial condition files are now published on zenodo (https://doi.org/10.5281/zenodo.6011618).**

**Comment E.11:**

*Line 154: Clarify how to process the data*

**Response E.11:**

[revised manuscript text omitted]

**Comment E.24:**

*Results: I am not sure if the new parameterization give improvement of important variables. For example, I don't believe that the new parameterization gives peak timing of LAI. Please check figures and numbers if they support the conclusion and results.*

**Response E.24:**

We included new elements in Table 4 showing the added-value of crop and irrigation options on LAI simulation. See response R1.2 to Reviewer 1:

"Corn is the dominant crop type in the considered irrigated area in Nebraska (Zhang et al. 2020). While the satellite LAI observations present a peak at the end of July, the modelled LAI is plateauing in August (Fig. 4). Corn LAI observations at the field scale for various agricultural management conditions are showed in Boedhram et al. (2001). These data show that the modelled LAI plateau in August at LAI values of about 3.5 m2m-2 is realistic."

This will be added in Section 3.1.

---

## Referee Report (RR1)

This study provides a new flexible irrigation and crop scheme, in which several parameters of irrigation and crop phenology could be modified by users. This is very important for detecting irrigation-induced impacts as it considers the spatial heterogeneity of agricultural activities. For the main idea of study, I have no further comments, but I still have some minor comments on the structure, language and some contents. I strongly recommend the authors to polish the full text, as there are some repititions, unclarity and even grammar errors.

The number of lines is based on gmd-2021-332-ATC1.pdf

L50: after 'Affect non-irrigated areas' cite de Vrese paper: Asian irrigation, African rain: Remote impacts of irrigation

L71: after 'vegetation density' (add some citations here)

L74: 'there is' to 'there are'

L76: cite Jägermeyr paper 'Water savings potentials of irrigation systems: global simulation of processes and linkages'

L76: 'The type of irrigation is recognized to' to 'Different irrigation types vary in'

L85: 'follow' to 'reproduce'

L92: I would put the SURFEX and ISBA part to model description in Section2. Here you should briefly describe your objective, and the new features of the new irrigation scheme and what it could be used for.

L101: Also here the simulation part should be in simulations protocol.

L113: 'model run' to 'simulation'

Sec 2: General comments: Section2.1 is called Model implementation and evaluation, actually you talk about study area and simulation settings. I would rename it experimental design or something else, then move it between the description of your new irrigation scheme and data.

L123: I won't say implementation is made over this region, you could say the simulations and evaluation are conducted over this region.

L128: 'irrigation' to 'irrigation activities'

L130: You don't need to specify the forcing data here, as you should do it in Data. Here you can just say meteorological forcing data.

L138: 'nature types'? Aren't they crop types?

L146: Specify the reason why you only choose the area with more than 50%. I guess it is because this is off-line simulation, so there will be no non-local effects, right?

L159: give the formulas of Pearson's correlation and RMSD.

L219: 'In ISBA_ref simulations' Here I would avoid mentioning simulations, instead, I would say 'In the original crop scheme of the ISBA model'. Other information regarding the simulations could be moved to section experimental design mentioned in the general comments

L237: What do you mean 'based on', or you just want to express that you used SURFEX v8.1 to do the simulations?

Section 2.3: General comments:

It is very hard to compare your new scheme and the old scheme, as the old scheme is not presented in paper at all. I would add a brief description of the old scheme, including how irrigation is triggered and where the water is applied. I would split this section to two subsections: the original crop and irrigation scheme and the new one, which would be more clear for readers.

L261: How the values of these parameters are decided? By observation or by calibration. Please specify.

L272: 'it is determined that' to 'The model determines …'

L275: use 'First, Second, …' or 'Firstly, Secondly, …'; 'it is checked that' to 'the model checks'

L280-284: I would just say that the model provides the options to …

L298: top layers? I would say total soil layers in root zone (if I understand correctly).

L301: Could you please specify the reason why this threshold decreases for new seasons?

L311: I would say 'the irrigation water can be intercepted by vegetation canopy.'

L448: why did you choose these two weather stations, one in irrigated land and one in rainfed irrigated land? Is it on purpose or just based on the availability?

L493: the first paragraph is unnecessary.

L512: I think taking no water availability into account could explain the slight overestimation.

L516: 'While … is realistic.' I don't think I understand this sentence. Satellite Lai observations are not realistic?

L557: Based on my understanding, you want to say that surrounded rainfed vegetation affects the phenology? It is better to specify it.

L561: 'Positive …' I would move it to the caption of the figure.

L646: I am wondering if it is possible to add more parameters related to phenology, like the growing period, peak dates, ect… I think this could be a good way of further improving the model performance.

L656: I would describe the poor representation of the cold season processes in ISBA and clarify why it could be the reason.

L685: What are the impacts of irrigation on atmospheric model simulations. I would go deeper how it may limit your study.

Sec.5: You didn't really describe what this study really implies. I would talk more about the advantage of the new irrigation scheme and for what it could be used, and some implications of this study. Example1. This flexible crop and irrigation scheme could take the spatial heterogeneity of irrigation activities into account, thus it is a better tool to detect irrigation-induced impacts on earth system. Example2. Results show that phenology parameters could modify the seasonal pattern of LAI and other variables, and irrigation could affect the magnitude of the variables. This could provide the information for further development.

Sup: Check the tense consistency you used in manuscript and supplements. Both present or past tense are ok but keep it consistent.

---

## Referee Report (RR2)

This study introduces the new improvements of a land surface model (LSM) ISBA regarding agricultural management. The authors explicitly explain the key mechanisms and associated parameterizations of the new scheme. And the performances of the new version is validated over an irrigated area in Nebraska, US. By comparing simulated eco-hydrological components to observations, the authors demonstrate the advantages of the new scheme in reproducing crop dynamics as well as hydrological processes.

Indeed, simulating crops and irrigation in LSMs is challenging, because it must obey associated biophysical mechanisms and adapt to the complexity of diverse cultivations. It is impressive that the authors made a significant contribution and provided a comprehensive validation from various perspectives. I can see a lot of implications based on this work. Its contribution to the model development is worth being published in GMD. However, although the manuscript has improved a lot after a major revision, I think a substantial revision is still required before acceptance.

Major comments:
1. Although the title mentions a new crop scheme, I didn't see a explicit introduction to it in the main text. Moreover, there is no independent validation of the new crop scheme (i.e., correlation coefficients and RMSD between ISBA_pheno and ISBA_ref). Moreover, it is trivial to mention SURFEX_v8.1. I suggest to enrich the contents for the new crop scheme only (details see some comments below) and remove SURFEX_v8.1 from the title.

2. The manuscript is too long for readers to get the key innovations.
    1) Sect 2.1: It is not necessary to introduce the details of the SURFEX platform. And please try to introduce something directly. For example, it is not necessary to mention ECOCLIMAP-II when this study used ECOCLIMAP-SG. Moreover, this sentence should appear in the section of experiment setting up.
    2) There are a lot of overlaps between section 2.2 and 2.2.1. And it is still unclear for me the key differences between the old and the new scheme. Please first simply introduce the key processes implemented by the former crop and irrigation schemes without any details (i.e., parameters, formula of SWI, etc) and emphasis the key limitations of them which will be solved by the new scheme.
    3) Sect 2.2.2: Although the aggregation rule is new, its key aim is to reduce the computing burden. Thus I suggest to simply introduce the aggregation process here and move the details to the supplementary materials. Moreover, the last paragraph talks about the water supply (unlimited) for irrigation. It may belong to Sect. 2.3.
    4) If authors underline the difference of metrics values rather than their spatial patterns, some figures can be simplified (optional).  a. Figure 6, 8, and 9: This three can be assembled together by combining their top panels together and a box plot (or a violin plot) aside instead of two associated images.  b. Figure 7, 10, and 12: They can be assembled together by making three subplots with time series of boxes or violins.
    5) Supplement S5 is not needed.

3. Two issues regarding validation must be dug out. Firstly, irrigation amount is highly influenced by precipitation, which however is poorly reproduced in all reanalysis data. I'm not surprise that the fairly performance of ERA5 precipitation (Fig. S4), which may lead to vast bias between simulated irrigation and census data. Thus, I suggest to show how simulated irrigation amount is improved by the new scheme in comparison to that based on the old scheme rather than to demonstrate the fit between simulation and census data.

Secondly, there are significant mismatches among ET products. Although assimilated with a lot of observations, the ET from GLEAM is a model output, not observations. The model have a very coarse representation of plant functional types, and irrigation processes is not taken into account (at least for v3.2b). So I don't think it is suitable to call it 'observations' (e.g., in Figure 9). Furthermore, it is not suitable to validate the simulated ET by GLEAM. Nothing can be demonstrated if the simulation is compared to an unreliable product. I suggest to remove this part. Other parts are sufficient to demonstrate the advances of the new scheme.

4. Line 49-50: There is still a large uncertain in terms of ET-precipitation feedback. It could either positive (rain prefers wet soil) or negative feedback (rain prefers dry soil), which relies on

numerous factors (e.g., surface heterogeneity, atmospheric boundary conditions, wind speed/ direction, spatial scales, etc). So I suggest to underline the contribution of irrigation to ET-precipitation feedback but avoid mentioning where the precipitation may occur.

5. Line 67-70: It is not true. See some new works including specific crop types, cultivation schedules, as well as multiple irrigation techniques.
Leng, G.Y., Leung, L.R., Huang, M.Y., 2017. Significant impacts of irrigation water sources and methods on modeling irrigation effects in the <scp>ACME</scp> <scp>L</scp> and Model. J. Adv. Model. Earth Syst. 9, 1665–1683. https://doi.org/10.1002/2016MS000885

Yin, Z., Wang, X.H., Ottlé, C., Zhou, F., Guimberteau, M., Polcher, J., Peng, S.S., Piao, S.L., Li, L., Bo, Y., Chen, X.L., Zhou, X.D., Kim, H., Ciais, P., 2020. Improvement of the Irrigation Scheme in the ORCHIDEE Land Surface Model and Impacts of Irrigation on Regional Water Budgets Over China. J. Adv. Model. Earth Syst. 12, 1–20. https://doi.org/10.1029/2019MS001770

6. Line 400-402: This is a literature at 2001. I guess both seed selection and fertilization contributes to the LAI increase. Figure 5b shows that the observed LAI in 2002 coincides with Boedhram et al. (2001). Is there an increasing trend of LAI?

Minor comments:

1. Line 36-37: It is not necessary to mention a specific region. The previous sentence already well describe the key drivers of increasing water demand.

2. Line 40: 'controlling' -> 'mitigating'.

3. Line 100: 'the' -> 'a'.

4. Line 104: Modify it to 'by driven by atmospheric forcing ..'

5. Line 149: Modify it to 'tends to optimize water withdrawal according to water extracting abilities of crops at different stages.'

6. Line 151: Modify it to 'sum of irrigated water will be validated by the ...'

7. Line 161-162: This sentence is abrupt. Consider to remove it.

8. Line 163-164: Please avoid hand-waving if there is little information about flood and drip irrigation in the manuscript.

9. Line 165: Seems Figure 1 doesn't appear in the previous content. Consider to change the order between Figure 1 and Figure 2.

10: Line 169: 'that' -> 'whether'.

11: Line 173-174: Remove this sentence.

12: Line 188: 'with no' -> 'without'.

13: Line 1001: blue dots?

14: Line 1029: For the correlation coefficient, it will be good enough if the p-value < 0.05.

---

## Author Response (AR2)

**Reviewer #1**

**Comment R1.1:**

This study provides a new flexible irrigation and crop scheme, in which several parameters of irrigation and crop phenology could be modified by users. This is very important for detecting irrigation-induced impacts as it considers the spatial heterogeneity of agricultural activities. For the main idea of study, I have no further comments, but I still have some minor comments on the structure, language and some contents. I strongly recommend the authors to polish the full text, as there are some repititions, unclarity and even grammar errors.

**Response R1.1:**

**Many thanks for your new inputs. We did our best to revise the English. In particular a number of typos you mentioned were corrected.**

**Comment R1.2:**

gmd-2021-332-ATC1.pdf L50: after 'Affect non-irrigated areas' cite de Vrese paper: Asian irrigation, African rain: Remote impacts of irrigation :

**Response R1.2:**

**This citation was added:**
**de Vrese, P., Hagemann, S. and Claussen, M.: Asian irrigation, African rain: Remote impacts of irrigation, Geophys. Res. Lett., 43, 3737-3745, https://doi.org/10.1002/2016GL068146, 2016.**

**Comment R1.3:**

L71: after 'vegetation density' (add some citations here)

**Response R1.3:**

**These two citations were added:**
**Perry, C.: Efficient irrigation; Inefficient communication; Flawed recommendations, Irrig. Drain., 56, 367–378, 2007.**
**Perry, C., Steduto, P., Allen, R. G., and Burt, C. M.: Increasing productivity in irrigated agriculture: Agronomic constraints and hydrological realities, Agr. Water Manage., 96, 1517–1524, 2009.**

**Comment R1.4:**

L76: cite Jägermeyr paper 'Water savings potentials of irrigation systems: global simulation of processes and linkages'.

**Response R1.4:**

**This citation was added:**

Jägermeyr, J., Gerten, D., Heinke, J., Schaphoff, S., Kummu, M., and Lucht, W.: Water savings potentials of irrigation systems: global simulation of processes and linkages, Hydrol. Earth Syst. Sci., 19, 3073–3091, https://doi.org/10.5194/hess-19-3073-2015, 2015.

**Comment R1.5:**

L92: I would put the SURFEX and ISBA part to model description in Section2. Here you should briefly describe your objective, and the new features of the new irrigation scheme and what it could be used for.

**Response R1.5:**

These 10 lines were moved to new Section 2.1.

**Comment R1.6:**

L101: Also here the simulation part should be in simulations protocol.

**Response R1.6:**

These 4 lines were moved to new section 2.2.2.

**Comment R1.7:**

Sec 2: General comments: Section2.1 is called Model implementation and evaluation, actually you talk about study area and simulation settings. I would rename it experimental design or something else, then move it between the description of your new irrigation scheme and data.

**Response R1.7:**

This section was renamed and moved to new Section 2.3 (Experimental design).

**Comment R1.8:**

L123: I won't say implementation is made over this region, you could say the simulations and evaluation are conducted over this region.

**Response R1.8:**

"implementation" was replaced by "simulations".

**Comment R1.9:**

L130: You don't need to specify the forcing data here, as you should do it in Data. Here you can just say meteorological forcing data.

**Response R1.9:**

This sentence was reworded as:
"The ISBA LSM simulations are made at a spatial resolution of 0.25° × 0.25°, over a 40-year period from 1979 to 2018. "

**Comment R1.10:**

L138: 'nature types'? Aren't they crop types?

**Response R1.10:**

Yes. "C3" was replaced by "C3 crops"

**Comment R1.11:**

L146: Specify the reason why you only choose the area with more than 50%. I guess it is because this is off-line simulation, so there will be no non-local effects, right?

**Response R1.11:**

The reason why we choose areas with more than 50 % irrigated surfaces is because we use offline simulations and that we want to focus on local effects irrigation.

The following sentence was added in new Section 2.3:

"For the intercomparison of the simulations we select areas where the irrigation fractional coverage is larger than 50 % as determined from the irrigation map."

was replaced by

"For the intercomparison of the simulations we select areas where the irrigation fractional coverage is larger than 50 % as determined from the irrigation map, in order to better assess the local effects of irrigation in offline simulations."

**Comment R1.12:**

L159: give the formulas of Pearson's correlation and RMSD.

**Response R1.12:**

We added Eqs (2-4) at the end of new section 2.3.

**Comment R1.13:**

L219: 'In ISBA_ref simulations' Here I would avoid mentioning simulations, instead, I would say 'In the original crop scheme of the ISBA model'. Other information regarding the simulations could be moved to section experimental design mentioned in the general comments.

**Response R1.13:**

We now say : "In this tudy".

**Comment R1.14:**

L237: What do you mean 'based on', or you just want to express that you used SURFEX v8.1 to do the simulations?

**Response R1.14:**

This sentence was reworded as:
"The SURFEX v8.1 version (Le Moigne et al., 2018) was used to do the simulations"

**Comment R1.15:**

Section 2.3. It is very hard to compare your new scheme and the old scheme, as the old scheme is not presented in paper at all. I would add a brief description of the old scheme, including how irrigation is triggered and where the water is applied. I would split this section to two subsections: the original crop and irrigation scheme and the new one, which would be more clear for readers.

**Response R1.15:**

The "Irrigation processes" subsection was renamed as "New irrigation processes" and the description of pre-existing elements of the model was moved before this subsection.

**Comment R1.16:**

L261: How the values of these parameters are decided? By observation or by calibration. Please specify.

**Response R1.16:**

These values are based on past studies. This sentence was replaced by:
"Using these values allows the model to predict a realistic amount of irrigation water over irrigated corn in southern France (Bonnemort et al., 1996; Voirin-Morel, 2003; Calvet et al., 2008)."
and a new reference was added:
Bonnemort, C., Bouthier, A., Deumier, J.-M., and Specty, R.: Conduire l'irrigation avec Irritel ; intérêts et limites, La Météorologie, 14, 36-43, https://doi.org/10.4267/2042/51182, 1996.

"This irrigation strategy tends to limit water applications when the plant is able to extract water from the soil." was added to new section 2.2.

**Comment R1.17:**

L280-284: I would just say that the model provides the options to …

**Response R1.17:**

"This new crop and irrigation scheme is able to"
was replaced by
"The new crop and irrigation scheme provides the option to".

**Comment R1.18:**

L298: top layers? I would say total soil layers in root zone (if I understand correctly).

**Response R1.18:**

"the number of to soil layers containing roots"
was replaced by
"the total number of soil layers in the root zone".

**Comment R1.19:**

L301: Could you please specify the reason why this threshold decreases for new seasons?

**Response R1.19:**

**The following sentence was added:**
**"This irrigation strategy tends to limit water applications when the plant is able to extract water from the soil. "**

**Comment R1.20:**

L311: I would say 'the irrigation water can be intercepted by vegetation canopy.'

**Response R1.20:**

**The following sentence was added:**
**"The irrigation water can be intercepted by vegetation canopy."**

**Comment R1.21:**

L448: why did you choose these two weather stations, one in irrigated land and one in rainfed irrigated land? Is it on purpose or just based on the availability?

**Response R1.21:**

**The sentence was reworded as: "The two weather stations are within 170 km of each other and correspond to contrasting environmental conditions."**

**Comment R1.22:**

L493: the first paragraph is unnecessary.

**Response R1.22:**

**Was deleted and replaced by "The results presented below are focused on the impacts of the crop phenology and irrigation implementation on the simulated land surface variables over Nebraska."**

**Comment R1.23:**

L512: I think taking no water availability into account could explain the slight overestimation.

**Response R1.23:**

**Yes. These sentences were rephrased as:**
**"The mean simulated value of the yearly irrigation water amount used for irrigation (271±75 mm year$^{-1}$) slightly overestimates the observed one (264±65 mm year$^{-1}$), with a difference of +2.7%. This difference could be explained by the availability of the water resource, not accounted for by the model yet."**

**Comment R1.24:**

L516: 'While … is realistic.' I don't think I understand this sentence. Satellite Lai observations are not realistic?

**Response R1.24:**

**This sentence was reworded as:**
**"The data from Boedhram et al. (2001) show that the modelled LAI plateau in August at LAI values of about 3.5 m$^2$m$^{-2}$ is realistic for irrigated corn. The satellite LAI observations are sensitive to both rainfed and irrigated vegetation. "**

**Comment R1.25:**

L557: Based on my understanding, you want to say that surrounded rainfed vegetation affects the phenology? It is better to specify it.

**Response R1.25:**

**This sentence was reworded as:**
**"Compared to crop simulations, the experiments with crop phenology (ISBA_pheno and ISBA_pheno_irr) present earlier peak LAI dates, because rainfed vegetation affects the phenology".**

**Comment R1.26:**

L561: 'Positive …' I would move it to the caption of the figure.

**Response R1.26:**

**Done.**

**Comment R1.27:**

L646: I am wondering if it is possible to add more paramets related to phenology, like the growing period, peak dates, ect… I think this could be a good way of further improving the model performance.

**Response R1.27:**

**The following sentence was added at the beginning of Section 4.1:**
**"The crop phenology model is very simple and adding more parameters related to phenology could be a way to further improve the model performance. Integrating satellite LAI observations using data assimilation could also be an option (Mucia et al., 2020)."**

**Comment R1.28:**

L656: I would describe the poor representation of the cold season processes in ISBA and clarify why it could be the reason.

**Response R1.28:**

**This sentence was reworded as:**
**"Moreover, the representation of the cold season processes is not perfect in ISBA (Decharme et al. 2019) and the model tends to underestimate snow depth and the length of the snow season."**

**Comment R1.29:**

L685: What are the impacts of irrigation on atmospheric model simulations. I would go deeper how it may limit your study.

**Response R1.29:**

**The following paragraph was added:**

"Over Nebraska, Szilagyi and Franz (2020) show that the decadal increase in irrigated land tends to trigger a reduction in precipitation over the most densely irrigated areas, of about -10 mm per decade. The largest precipitation suppression is observed at Spring, in March, before the corn growing season, in relation to larger soil water content values. In our simulations, ISBA_pheno_irr presents larger soil moisture values than ISBA-ref in March (see Fig. S3.1), but this is mainly due to crop phenology."
Reference:
Szilagyi, J., Franz, T.E.: Anthropogenic hydrometeorological changes at a regional scale: observed irrigation–precipitation feedback (1979–2015) in Nebraska, USA, Sustain. Water Resour. Manag. 6, 10 pp., https://doi.org/10.1007/s40899-020-00368-w, 2020.

**Comment R1.30:**

Sec.5: You didn't really describe what this study really implies. I would talk more about the advantage of the new irrigation scheme and for what it could be used, and some implications of this study. Example1. This flexible crop and irrigation scheme could take the spatial heterogeneity of irrigation activities into account, thus it is a better tool to detect irrigation-induced impacts on earth system. Example2. Results show that phenology parameters could modify the seasonal pattern of LAI and other variables, and irrigation could affect the magnitude of the variables. This could provide the information for further development.

**Response R1.30:**

Many thanks for this suggestion. We added the following paragraph at the end of the Conclusion:

"This flexible crop phenology and irrigation scheme could take the spatial heterogeneity of irrigation activities into account, and detect irrigation-induced impacts on Earth system simulations. Our results show that crop phenology parameters could modify the seasonal pattern of the simulation of LAI, soil moisture, evapotranspiration and plant carbon uptake, and that irrigation could affect their magnitude. This could provide the basis for further development in offline and online applications of the ISBA model."

**Comment R1.31:**

Sup: Check the tense consistency you used in manuscript and supplements. Both present or past tense are ok but keep it consistent

**Response R1.31:**

Thanks for noting this. We moved all past tense sentences to present tense.

**Reviewer #2**

**Many thanks for your comments. They are addressed below. Also, a number of typos you mentioned were corrected.**

**Comment R2.1:**

Abstract L 24: "The ISBA simulations with and without irrigation scheme…" Is this conclusive as the comparisons are also made for simulations with/without crop phenology?

**Response R2.1:**

**"The ISBA simulations with and without the irrigation scheme are compared to different satellite-based observations. "**
**was replaced by**
**"The ISBA simulations with and without the new crop phenology and irrigation scheme are compared to different satellite-based observations. "**

**Comment R2.2:**

L105 – 110: may need to modify the description of the scope of this study as you also changed the title by including the role of crop phenology.

**Response R2.2:**

**"Section 2 presents the observational datasets, the current version of the ISBA LSM, the description of the new crop and irrigation scheme, followed by a description of the validation protocol. Section 3 illustrates the impact of the new scheme when compared to a model run without irrigation. An evaluation of the performance of the model is made over Nebraska. Section 4 discusses the added value and the limits of the newly implemented irrigation scheme. Finally, section 5 presents the conclusions and future research directions."**
**was replaced by :**
**"Section 2 presents a description of the ISBA LSM, the new crop and irrigation scheme, the validation protocol, followed by a description of the observational datasets. Section 3 illustrates the impact of the new scheme when compared to simulations without crop phenology and without irrigation. An evaluation of the performance of the model is made over Nebraska. Section 4 discusses the added value and the limits of the newly implemented irrigation scheme. Finally, section 5 presents the conclusions and future research directions."**

**Comment R2.3:**

Figure 1: Please add the latitude and longitude coordinates for the maps.

**Response R2.3:**

**Thanks for noting this. Figure 1 was updated and coordinates are now more visible.**

**Comment R2.4:**

L136: Why only mentioned the LAI comparison here?

**Response R2.4:**

**The following sentence was added:**
**"In addition to LAI, other variables are considered: gross primary production, evapotranspiration and land surface temperature."**

**Comment R2.5:**

What is the spatial resolution of the model simulations? Please clarify in section 2.

**Response R2.5:**

**The sentence is now in Section 2.3 and was reworded as:**
**"The ISBA LSM simulations are made at a spatial resolution of 0.25° × 0.25°, over a 40-year period from 1979 to 2018. "**

**Comment R2.6:**

L176: Is there any sensitivity analysis done regarding the magnitude of this parameter (30mm)? Similarly, any sensitivity test conducted for the choice of SWI threshold? How sensitive your result would be to these irrigation parameters? And how does this sensitivity transfer to vegetation conditions?

**Response R2.6:**

**These values are based on past studies. Using these values allows the model to predict a realistic amount of irrigation water over irrigated maize in southern France (Bonnemort et al., 1996; Voirin-Morel, 2003; Calvet et al., 2008). This irrigation strategy tends to limit water applications when the plant is able to extract water from the soil.**
**This sentence was replaced by:**
**"Using these values allows the model to predict a realistic amount of irrigation water over irrigated corn in southern France (Bonnemort et al., 1996; Voirin-Morel, 2003; Calvet et al., 2008). "**
**and a new reference was added:**
**Bonnemort, C., Bouthier, A., Deumier, J.-M., and Specty, R.: Conduire l'irrigation avec Irritel ; intérêts et limites, La Météorologie, 14, 36-43, https://doi.org/10.4267/2042/51182, 1996.**

**Comment R2.7:**

L197: "Irrigation can optionally be triggered without considering…" It is a bit hard to understand this as you mentioned earlier that the irrigation time window is based on the emergence and harvest dates.

**Response R2.7:**

**This sentence was reworded as:**
**"Irrigation can optionally be triggered without considering any specific crop phenology parameter but this option was not considered in this study."**

**Comment R2.8:**

Why don't the emergence and harvest dates differ among different vegetation types? Is this assumption reasonable? If mentioned as a limitation, could you populate the discussion on how this assumption may affect the states and fluxes?

**Response R2.8:**

This information is now given in the new 2.3 section "Experimental design", after the model has been presented, as well as the model parameters. In this study, the main irrigated vegetation type is corn. This is indicated now. The discussion of Section 4.1 was completed with the following senstence:
"The crop phenology model is very simple and adding more parameters related to phenology could be a way to further improve the model performance. Integrating satellite LAI observations in ISBA using sequential data assimilation is also an option (Mucia et al., 2020)."

**Comment R2.9:**

Line 300: Is the model simulation conducted at a resolution of 300m * 300m? It is not quite clear why the irrigation map needs to be transferred to 300m*300m if the model resolution is not at this scale.

**Response R2.9:**

We achnowledge that this is confusing. This sentence was moved to another part of the new section 2.2.

**Comment R2.10:**

L343-345: "It is available every 10 days…does not cover the whole simulation time period (1979 to 2018)" These sentences contain contradictory information, please clarify.

**Response R2.10:**

Thanks for noting this.
"It is available every 10 days for all simulation years. The LAI time series is available from 1999 onward. It does not cover the whole simulation time period (1979 to 2018)."
was replaced by
"It is available every 10 days from 1999 onward. It does not cover the whole simulation time period (1979 to 2018).".

**Comment R2.11:**

I don't think GLEAM ET is a good candidate reference datasets as the model does not explicitly include the irrigation signal. Please consider select other ET reference datasets that can detect the irrigation signal or include several widely used ET datasets for comparison besides GLEAM.

**Response R2.11:**

See Response R2.15.

**Comment R2.12:**

Figure 4: I would suggest converting the irrigation number to total irrigation amount in this figure. Your y-axis is irrigation number while some of the data shown are compared in terms of amount.

**Response R2.12:**

**This is a very good point. Irrigation number was converted to Irrigation amount in Fig. 4.**

**Comment R2.13:**

Section 3.1: Other than the averaged irrigation amount, could you comment on the model performance in simulating irrigation amount in wet/dry years? Even though it is not possible to evaluate the interannual variation as the data is only available every 5 years, it is possible to at least see how the model perform in wet/dry conditions.

**Response R2.13:**

**Yes. We revised Fig. 4 in order to indicate the 2000 and 2005 dry years, and the 2010 wet year. The last paragraph of Section 3.1 was replaced by**
**"This difference could be explained by the availability of the water resource, not accounted for by the model yet. The large observed irrigation amounts in 2000 and 2005, larger than 300 mm year$^{-1}$, are relatively well represented by the model. On the other hand, the observed small irrigation amount for the 2010 wet year, is overestimated by about 110 mm year$^{-1}$."**

**Comment R2.14:**

Figure 6&7: I would suggest including correlation and RMSD difference between ISBA_pheno and ISBA_ref and then discuss the major contributor for this improvement. It seems that irrigation only plays an additive role in improving the vegetation seasonal cycle as compared to the role of including crop phenology.

**Response R2.14:**

**We checked this. Actually, ISBA_pheno correlation and RMSD differences with respect to ISBA_ref are nearly identical to those showed for ISBA_pheno_irr in Figs. 6-7.**
**The following sentence was added at the end of Section 3.3:**
**"The ISBA_pheno correlation and RMSD differences with respect to ISBA_ref are nearly identical to those showed for ISBA_pheno_irr in Figs. 6-7 (not shown)."**
**In Section 4.1,**
**"The results of our numerical experiments over Nebraska show that considering crop phenology and irrigation improves the consistency of the simulations with LAI and GPP observations. The corresponding correlation and RMSD scores are improved. Two new developments can explain this behaviour: (1) the crop phenology parameters used to force emergence and harvest dates reduce the length of the growing season, delay spring growth and avoid a regrowth in the autumn, and (2) the irrigation limits the water stress and enhances plant growth at summertime. Nevertheless they both have shortcomings and their performance could be limited by difficulties in simulating processes that are not directly related to irrigation."**
**was replaced by**

"The results of our numerical experiments over Nebraska show that considering crop phenology improves the consistency of the simulations with LAI and GPP observations. The corresponding correlation and RMSD scores are improved. The crop phenology parameters used to force emergence and harvest dates reduce the length of the growing season, delay spring growth and avoid a regrowth in the autumn. It seems that irrigation only plays an additive role in improving the vegetation seasonal cycle as compared to the role of including crop phenology (Section 3.3). Both crop phenology and irrigation models have shortcomings and their performance could be limited by difficulties in simulating processes that are not directly related to irrigation."

**Comment R2.15:**

Figure 9: Degradation in R for ET for most of the months considered may be evidence that GLEAM is not suitable to be referred for irrigation impacted ET comparison. Please investigate other reference datasets. Besides, ET products are subject to uncertainties in magnitude and RMSD needs to be discussed under this context.

**Response R2.15:**

Yes. We added additional results in the Supplement (Table S3.1, Table S3.2, Figure S3.5).
The following sentence was added at the end of Section 3.5:
"Degradation in *r* can be observed at some locations throughout the growing season. "
The following paragraph was added at the end of a new Section 4.2:
"Finally, degradation in r for evapotranspiration in Fig. 10 may be evidence that GLEAM may not be considered as a suitable reference for evapotranspiration comparisons in areas impacted by irrigation. The use of other datasets is investigated in Supplement 3. In particular, in situ observations over an irrigated corn field (Suyker and Verma, 2009) are used. Table S3.1 shows that GLEAM tends to underestimate evapotranspiration by 20 % during the growing season (from May to September). During the non-growing season, the ISBA_pheno_irr model overestimates evapotranspiration by 48 %. Table S3.2 and Fig. S3.5 show that the ISBA_pheno_irr evapotranspiration peak (in June) tends to happen too early. Mean values of near-surface wind speed are particularly large over Nebraska, especially at wintertime and springtime (Chen, 2020). This feature could exacerbate the impact of a misrepresentation of soil evaporation."

**Reference:**

Chen, L.: Impacts of climate change on wind resources over North America based on NA-CORDEX, Renewable Energy, 153, 1428-1438, https://doi.org/10.1016/j.renene.2020.02.090, 2020.

**Comment R2.16:**

For all the correlation related analysis, how the values are selected to mask out the grid cells with white color? It would be better to add a significance test and use the significance result to mask out the grid cells.

**Response R2.16:**

We tried to improve the readability (also for color-blind readers) of Figures 6 to 12 by simplifying the color bars and adding white plus symbols for areas presenting the largest improvements.

**Comment R2.17:**

L521: Is it possible to derive emergence and harvest dates based on the LAI observation so that it can represent the interannual variation? Please comment on this.

**Response R2.17:**

**The following sentence was added in Section 4.1:**
**"Also, emergence and harvest dates could be derived from the LAI observation in order to better represent the interannual variation."**

**Comment R2.18:**

Please make sure all the supplemental materials are mentioned in the main manuscript to support relevant statements.

**Response R2.18:**

**We checked that and we added a few more citations to the supplemental materials.**

---

## Author Response (AR3)

**Reviewer #1**

**Comment R1.1:**

It is impressive that the authors made a significant contribution and provided a comprehensive validation from various perspectives. I can see a lot of implications based on this work. Its contribution to the model development is worth being published in GMD. However, although the manuscript has improved a lot after a major revision, I think a substantial revision is still required before acceptance.

**Response R1.1:**

**Many thanks for these positive comments and for your in-depth review of the manuscript. We did our best to further revise the manuscript.**

**Comment R1.2:**

Although the title mentions a new crop scheme, I didn't see a explicit introduction to it in the main text. Moreover, there is no independent validation of the new crop scheme (i.e., correlation coefficients and RMSD between ISBA_pheno and ISBA_ref). Moreover, it is trivial to mention SURFEX_v8.1. I suggest to enrich the contents for the new crop scheme only (details see some comments below) and remove SURFEX_v8.1 from the title.

**Response R1.2:**

**We agree that the way we presented ISBA_pheno results could give the wrong impression that the new crop phenology scheme results are not presented. The title could actually be improved because the meaning of « crop scheme » is not clear. The manuscript already contains specific results for the « ISBA_pheno » experiment. Differences between ISBA_ref, ISBA_pheno, and ISBA_pheno_irr are shown in Figures 5, 6, 8, 9, 11, S3.5, and in Table 4. Score maps in Figures 6-10 and 12 are shown for ISBA_pheno_irr only because the score maps are nearly identical as for ISBA_pheno. This is indicated at the end of section 3.3: 'The ISBA_pheno correlation and RMSD differences with respect to ISBA_ref are nearly identical to those showed for ISBA_pheno_irr in Figs. 6-7 (not shown).' Moreover, quantitative differences in the performance of ISBA_pheno and ISBA_pheno_irr are presented in Sections 3.2 and 3.4.**
**In order to further clarify this, we:**
**- (1) changed the title from**
**'Implementation of a new crop and irrigation scheme in the ISBA land surface model using SURFEX_v8.1'**
**to**
**'Implementation of a new crop phenology and irrigation scheme in the ISBA land surface model using SURFEX_v8.1'.**
**In order to be consistent with the title, 'crop and irrigation scheme' was replaced by 'crop phenology and irrigation scheme' everywhere in the text. Including « SURFEX_v8.1 » in the title was requested by the executive Editor. This is needed to follow GMD guidelines. The paper was submitted to the SURFEX special issue.**
**- (2) created a new subsection in Section 2.2, dedicated to 'new crop phenology processes'.**

**Comment R1.3:**

Sect 2.1: It is not necessary to introduce the details of the SURFEX platform. And please try to introduce something directly. For example, it is not necessary to mention ECOCLIMAP-II when this study used ECOCLIMAP-SG. Moreover, this sentence should appear in the section of experiment setting up.

**Response R1.3:**

We agree. We deleted part of the text of Section 2.1 and moved part of it to Sections 2.2.1 and 2.3. Since this paper is for the GMD SURFEX special issue, we need to keep a brief description of the SURFEX framework.

**Comment R1.4:**

There are a lot of overlaps between section 2.2 and 2.2.1. And it is still unclear for me the key differences between the old and the new scheme. Please first simply introduce the key processes implemented by the former crop and irrigation schemes without any details (i.e., parameters, formula of SWI, etc) and emphasis the key limitations of them which will be solved by the new scheme.

**Response R1.4:**

We created a new subsection in Section 2.2, dedicated to 'new crop phenology processes' (see Response 1.2) and we slightly reduced the old section 2.2.1. The site-level processes represented by the new irrigation scheme are not fundamentally different from the old one. The novelty is that the new scheme is able to spatialize irrigation, to manage harvest dates, to handle several irrigations types over several vegetation types, and is interoperable with the most recent versions of the soil water diffusion model. A lot of technical work was done to achieve this result. In order to clarify this
'In this study, a pre-existing simple irrigation scheme (Calvet et al., 2008) within the ISBA LSM is upgraded to build a new version able to work at a global scale and to represent several types of irrigation practices' (Section 2.2)
was replaced by
'An old irrigation scheme working at a local scale (Calvet et al., 2008) is available in the ISBA LSM. Major limitations of the old scheme are the lack of (1) spatialization at a global scale, (2) representation of harvest, (3) diversity of irrigation types and irrigated vegetation types, (4) interoperability with the multi-layer soil hydrology scheme. Key processes implemented in this scheme are briefly described below.'.
Moreover, 'The yearly sum of irrigated water can be compared to the USGS data described in Section 2.4.3. The irrigation water flux is evenly distributed over a period of time of 8 hours (by default) and is applied on top of the vegetation canopy like precipitation. The irrigation water can be intercepted by vegetation canopy.'
was moved to new Section 2.2.2.

**Comment R1.5:**

Sect 2.2.2: Although the aggregation rule is new, its key aim is to reduce the computing burden. Thus I suggest to simply introduce the aggregation process here and move the details to the supplementary materials. Moreover, the last paragraph talks about the water supply (unlimited) for irrigation. It may belong to Sect. 2.3.

**Response R1.5:**

**We agree. This part of the « new aggregation rules » section was moved to Supplement 1, together with Fig. 3. The last paragraph was moved to Section 2.3.**

**Comment R1.6:**

If authors underline the difference of metrics values rather than their spatial patterns, some figures can be simplified (optional). a. Figure 6, 8, and 9: This three can be assembled together by combining their top panels together and a box plot (or a violin plot) aside instead of two associated images. b. Figure 7, 10, and 12: They can be assembled together by making three subplots with time series of boxes or violins.

**Response R1.6:**

**Thanks for this suggestion to improve the readability of the paper. In response to comment R1.9, we removed Figures 9 and 10 related to GLEAM. We reorganized former Figs. 6 and 8. LAI and GPP time series are now in the same Figure (new Fig. 5). LAI and GPP score differences are now in the same Figure (new Fig. 6).**

**Comment R1.7:**

Supplement S5 is not needed.

**Response R1.7:**

**We believe that the technical Supplement 5 is needed by SURFEX users using the new scheme or willing to reproduce our experiments. Supplement 5 is cited in Section 2.2.**

**Comment R1.8:**

Two issues regarding validation must be dug out. Firstly, irrigation amount is highly influenced by precipitation, which however is poorly reproduced in all reanalysis data. I'm not surprise that the fairly performance of ERA5 precipitation (Fig. S4), which may lead to vast bias between simulated irrigation and census data. Thus, I suggest to show how simulated irrigation amount is improved by the new scheme in comparison to that based on the old scheme rather than to demonstrate the fit between simulation and census data.

**Response R1.8:**

**The precipitation quality issue is briefly discussed in Section 4.1. In response to Reviewer 2 (see response R2.3) we completed Supplement S4 to better capture the shortcomings of ERA5 precipitation.**

**Comment R1.9:**

Secondly, there are significant mismatches among ET products. Although assimilated with a lot of observations, the ET from GLEAM is a model output, not observations. The model have a very coarse representation of plant functional types, and irrigation processes is not taken into account (at least for v3.2b). So I don't think it is suitable to call it 'observations' (e.g., in Figure 9). Furthermore, it is not suitable to validate the simulated ET by GLEAM. Nothing can be demonstrated if the simulation is compared to an unreliable product. I suggest to remove this part. Other parts are sufficient to demonstrate the advances of the new scheme.

**Response R1.9:**

**We agree. We removed Section 3.5 and Figures 9 and 10 related to GLEAM. Section 4.2 was revised accordingly. This will further improve the readability of the paper.**

**Comment R1.10:**

Line 49-50: There is still a large uncertain in terms of ET-precipitation feedback. It could either positive (rain prefers wet soil) or negative feedback (rain prefers dry soil), which relies on numerous factors (e.g., surface heterogeneity, atmospheric boundary conditions, wind speed/ direction, spatial scales, etc). So I suggest to underline the contribution of irrigation to ETprecipitation feedback but avoid mentioning where the precipitation may occur.

**Response R1.10:**

**We agree.**

**'Water vapour originating from large scale irrigation water supply can be recycled to rainfall and affect non-irrigated areas'**

**was replaced by**

**'Water vapour originating from large scale irrigation water supply can be recycled to rainfall'.**

**Comment R1.11:**

Line 67-70: It is not true. See some new works including specific crop types, cultivation schedules, as well as multiple irrigation techniques.

Leng, G.Y., Leung, L.R., Huang, M.Y., 2017. Significant impacts of irrigation water sources and methods on modeling irrigation effects in the <scp>ACME</scp> <scp>L</scp> and Model. J. Adv. Model. Earth Syst. 9, 1665–1683. https://doi.org/10.1002/2016MS000885

Yin, Z., Wang, X.H., Ottlé, C., Zhou, F., Guimberteau, M., Polcher, J., Peng, S.S., Piao, S.L., Li, L., Bo, Y., Chen, X.L., Zhou, X.D., Kim, H., Ciais, P., 2020. Improvement of the Irrigation Scheme in the ORCHIDEE Land Surface Model and Impacts of Irrigation on Regional Water Budgets Over China. J. Adv. Model. Earth Syst. 12, 1–20. https://doi.org/10.1029/2019MS001770

**Response R1.11:**

**We replaced**

**'Efforts are made to achieve this goal in the Community Land Model (CLM) and Noah-MP LSMs (Felfelani et al. 2020, Zhang et al. 2020, respectively).'**

**by**

**'For example, efforts are made to achieve this goal in the Community Land Model (CLM), in Noah-MP, in Accelerated Climate Modeling for Energy (ACME), and in ORganizing Carbon and Hydrology in Dynamic EcosystEms (ORCHIDEE) LSMs (Felfelani et al. 2020, Zhang et al. 2020, Leng et al. 2017, and Yin et al. 2020, respectively).'.**

**Comment R1.12:**

Line 400-402: This is a literature at 2001. I guess both seed selection and fertilization contributes to the LAI increase. Figure 5b shows that the observed LAI in 2002 coincides with Boedhram et al. (2001). Is there an increasing trend of LAI?

**Response R1.12:**

**The 1999-2018 LAI observations used to produce this Figure tend to increase through time. The mean annual LAI presents a trend of 1.57 $10^{-2}$ $m^2 m^{-2} yr^{-1}$. The ISBA_pheno_irr simulation presents a less pronounced trend of 1.23 $10^{-2}$ $m^2 m^{-2} yr^{-1}$. The simulation accounts for the atmospheric $CO_2$ effect on plant growth and water use efficiency using the method descibed in Calvet et al. (2008) but this effect mainly concerns C3 plants. Agricultural practices such as seed selection, pest control and fertilization are not represented. This could explain the less pronounced trend in the simulation. Boedhram observations were made in 1994 and 1995 but the observed LAI trend is not large enough to question the use of these data.**

**Comment R1.13:**

Minor comments:
1. Line 36-37: It is not necessary to mention a specific region. The previous sentence already well describe the key drivers of increasing water demand.
2. Line 40: 'controlling' -> 'mitigating'.
3. Line 100: 'the' -> 'a'.
4. Line 104: Modify it to 'by driven by atmospheric forcing ..'
5. Line 149: Modify it to 'tends to optimize water withdrawal according to water extracting abilities
of crops at different stages.'
6. Line 151: Modify it to 'sum of irrigated water will be validated by the ...'
7. Line 161-162: This sentence is abrupt. Consider to remove it.
8. Line 163-164: Please avoid hand-waving if there is little information about flood and drip irrigation in the manuscript.
9. Line 165: Seems Figure 1 doesn't appear in the previous content. Consider to change the order
between Figure 1 and Figure 2.
10: Line 169: 'that' -> 'whether'.

11: Line 173-174: Remove this sentence.
12: Line 188: 'with no' -> 'without'.
13: Line 1001: blue dots?
14: Line 1029: For the correlation coefficient, it will be good enough if the p-value < 0.05.

**Response R1.13:**

**Many thanks. All your suggested changes were made. For p-value, see response R2.5.**

**Reviewer #2**

**Comment R2.1:**

The authors have done a commendable job in addressing most of the questions in the revised submission and the overall quality of the presentation has improved substantially. I have a few additional comments here and hope this helps.

**Response R2.1:**

**Many thanks for these positive comments and for your in-depth review of the manuscript. We did our best to further revise the manuscript.**

**Comment R2.2:**

Comment to response R2.5: a bit more detail is needed to argue the robustness of using a model resolution of 0.25 deg when other input datasets are available at a much finer resolution such as the irrigation map. I understand that the resolution of the ERA5 met-forcing might be a factor at play, but I think it would help by adding discussions on how the choice of this spatial resolution may affect resolving the smaller irrigation variabilities and whether this may affect the results under different conditions? It might be good to clarify this.

**Response R2.2:**

**We agree that the spatial resolution context needs to be clarified, although intercomparing several versions of the model at various spatial resolutions is out of the scope of this work.**

[revised manuscript text omitted]

**Comment R2.4:**

Comment to response R2.15. Thanks for conducting additional analysis on ET. Figure 5&6 indicates that LAI seasonality differs largely between ISBA_ref and ISBA_pheno_irr, however, when looking at the seasonal cycle for ET, the peak timing seems to be not much affected. I wonder how the ISBA model estimate ET? Is LAI or vegetation related parameter at play? I think it would be good to elaborate the ET discussion in terms of its response to LAI from both the modeling and observational aspect.

**Response R2.4:**

**Observational aspect: in response to Reviewer 1, we removed the ET comparison with GLEAM from the manuscript, since GLEAM has shortcomings over irrigated areas and cannot be considered as an observation. However, we kept Supplement 3 and a version of the 4.2 discussion section focussing on modeling aspects.**

**Modeling aspects: the following sentences were added at the beginning of Section 4.2: 'All evaporation terms (plant transpiration, soil evaporation, interception) are simulated by ISBA. Under given environmental conditions, the simulated plant transpiration is not proportional to LAI. A simple canopy radiative transfer model is used to simulate the available photosynthetically active radiation (PAR) within the vegetation canopy. The response of GPP and transpiration to PAR and to LAI is controlled by this radiative transfer model. Photosynthesis and transpiration are calculated for three layers and summed to calculate canopy level values. For large LAI values, the mean leaf-level GPP and transpiration simulations are reduced in relation to smaller vegetation transmittance to solar radiation. The impact of changes in LAI on mean leaf-level GPP and transpiration is large at intermediate LAI values ranging from 1 to 3 $m^2m^{-2}$. It is much reduced for LAI values larger than 3 $m^2m^{-2}$. An improved version of this radiative transfer model able to represent ten canopy layers, with a more realistic response to solar zenith angle will be available in the next version of SURFEX (Delire et al. 2020).'**

0.05. Note that for LST, correlation and RMSD differences were not significant and we decided to remove former Fig. 12 together with the paragraph citing Fig. 12.

In order to improve readability, we reorganized former Figs. 6 and 8. LAI and GPP time series are now in the same Figure (new Fig. 5). LAI and GPP score differences are now in the same Figure (new Fig. 6).

The following sentence was added at the end of Section 2.3:

'The significance of r, r differences, and RSMD differences is tested using Fisher's test, Fisher's z test, and paired sample Student's test, respectively. Significance levels of 0.01, 0.05, and 0.05 are used, respectively.'

---

## Author Response (AR4)

**Reviewer #4 (anonymous)**

**Comment R1.1:**

Thanks for the hard work, The manuscript has been substantially improved. And all my comments are well addressed by the authors. The innovation of this study and associated evidence are elegantly presented. However, I find two tiny issues that may be helpful. Thus I recommend considering publication on GMD after a minor revision.

**Response R1.1:**

**Many thanks for these positive comments.**

**Comment R1.2:**

Line 211-212: It will be great if the algorithm can be briefly introduced in one or two sentences.

**Response R1.2:**

**"In order to limit the computing time, vegetation types can (optionally) be gathered. In this case vegetation "patches" are created (see Supplement S1 and Fig. S1.4)."**

**was replaced by**

**"In order to limit the computing time, vegetation types can (optionally) be gathered. In this case vegetation "patches" are created (see Supplement S1 and Fig. S4). Firstly, irrigated nature types are duplicated in order to ensure the distinction of irrigated and rainfed soil water budgets. Patch aggregation rules are then used to merge the nature types. Finally, model parameter values are computed following the new patch fraction map."**

**Comment R1.3:**

I assume that supplementary documents are used to support related statements. Thus I suggest citing them rather than introducing them (e.g., Line 537-539 and 542-543).

**Response R1.3:**

**Thanks for this suggestion. We rephrased all the sentences that were directly referring to Supplement Figures. Note that numbers of Supplement Figures and Tables were changed.**

**Reviewer #5 (Fabian Stenzel)**

**Comment R2.1:**

The present study ("Implementation of a new crop phenology and irrigation scheme in the ISBA land surface model using SURFEX_v8.1") introduces a new phenology and irrigation scheme for the ISBA LSM and evaluates its performance against observational data from a densely irrigated region of Nebraska (USA). It becomes clear that the main improvement for better performance regarding LAI and GPP stems from the improved phenology with prescribed emergence and harvest dates. The irrigation scheme does not add much with respect to the aforementioned variables, but provides reasonable water use values with respect to observations. I have been only involved now where the manuscript is already in round 3 of revisions. Therefore I interpret my main responsibility is to judge whether the present version of the manuscript is fit for publication and the authors have taken care of all points raised by the two previous reviewers. This is the case. The only thing missing in terms of reproducibility of the study is a step-by-step explanation of how to use the data in the ZENODO archive(s) together with the SURFEX code to redo the simulations and any potential postprocessing scripts. I suggest to add a README file to the ZENODO archive containing this information. The manuscript itself is written very clearly and I enjoyed reading it. However, since I read the article for the first time and with fresh eyes, I noticed some minor things that could still be improved and I ask the authors to include them in the final version of their manuscript. I apologize for this, because I know that for authors, introducing new reviewers late in the review process is annoying, but at the same time I hope that they seize the opportunity to further increase the quality of their (already good) manuscript by including my remarks. I am looking forward to read about future work on the global evaluation of the phenology and irrigation schemes in ISBA.

**Response R2.1:**

**Many thanks for these positive comments. We did our best to further revise the manuscript.**

**Concerning the reproducibility of simulations from the ZENODO data, we have enhanced the explanation of how to install the specific version of the ISBA model, to download different forcing files. We added a specific configuration file for each simulation and the script used to launch them. We strived to make it more readable. As suggested, we added in the ZENODO archive a step by step explanation in a README.txt (https://doi.org/10.5281/zenodo.7221291). It is nevertheless important, as mentioned in the readme file, to note that ISBA is a complex land surface model able to work at a global scale within the SURFEX modeling platform. Some training is needed for new users of SURFEX. We included contact points for technical support.**

**Comment R2.2:**

What purpose is the model generally used for? One main purpose of crop models is to provide harvest amounts. Therefore I was wondering, why (additional to the irrigation water amount, LAI and GPP) you did not look at how harvests compared between the 3 model versions? I would suspect that here you might see a stronger difference between ISBA_pheno and ISBA_pheno_irr, suggesting that including irrigation is worthwhile.

**Response R2.2:**

**ISBA is not a crop model (see Response 2.3). However, ISBA is able to simulate the green above-ground biomass (AGB) and this quantity can be compared with grain yield after model calibration or after the assimilation of satellite-derived LAI products (Calvet et al. 2012, Dewaele et al. 2017).**
**A new subfigure was added to Fig. S9 (Fig. S9d):**

[Figure]

**The main difference between annual AGB peak values simulated by ISBA_pheno_irr and ISBA_pheno are observed in 2017, which is a relatively dry year in the ERA5 reanalysis. Growing season (May-September) accumulated precipitation amounts in ERA5 are equal to 520, 499, 339, and 578 mm from 2015 to 2018, respectively. Irrigation increases the peak AGB value by 6 % in 2017.**

**References:**

**Calvet, J.-C., Lafont, S., Cloppet, E., Souverain, F., Badeau, V., Le Bas, C., "Use of agricultural statistics to verify the interannual variability in land surface models: a case study over France with ISBA-A-gs", Geosci. Model Dev., 5, 37-54, https://doi.org/10.5194/gmd-5-37-2012, 2012.**

**Dewaele, H., Munier, S., Albergel, C., Planque, C., Laanaia, N., Carrer, D., and Calvet, J.-C.: Parameter optimisation for a better representation of drought by LSMs: inverse modelling vs. sequential data assimilation, Hydrol. Earth Syst. Sci., 21, 4861–4878, https://doi.org/10.5194/hess-21-4861-2017, 2017.**

**Comment R2.3:**

Lines 58-72: It it great that you implement irrigation into ISBA. However your are making it sound like just very few other models have implemented irrigation. I don't think that is a fair point, as basically all crop models have implemented it. Many of them can (depending on your definition) be regarded as a LSM.

**Response R2.3:**

**Many thanks for this comment. We completed the definition of a LSM and made clear that LSMs are not crop models.**

**'Land surface models (LSMs) represent land surface biophysical processes and variables, including soil moisture and vegetation biomass, in a way that is fully consistent with the representation of carbon, water and energy fluxes.'**

**was replaced by**

**'Land surface models (LSMs) provide lower boundary conditions to climate and weather forecast atmospheric models. The new generation of LSMs is able to represent land surface biophysical processes and variables, including soil moisture and vegetation biomass, in a way that is fully consistent with the representation of carbon, water and energy fluxes. LSMs differ from crop models in the sense that they do not explicitly represent all the agricultural practices, nor crop yields. While most crop models have implemented irrigation, irrigation is not represented by all LSMs.'**

**Comment R2.4:**

Lines 129-134: Your argument for having decreasing SWI thresholds for subsequent irrigation events is weak. The root fraction is already part of SWIroot_zone. Additionally: If the irrigation water amount is still 30mm for subsequent irrigation events triggered at lower SWI, you would need more water to fill the soil again, right?

**Response R2.4:**

**Yes, we tried to consolidate the argumentation. The idea behind this approach is that irrigation does not completely refill the soil, especially at the end of the growing season. Mechanical harvest requires relatively dry conditions to avoid soil compaction. The crop is allowed to use rainwater together with the initial available water content of the soil. Results presented in Section 3.1 show that this hypothesis is realistic for Nebraska.**

**'This irrigation strategy allows the optimization of water withdrawal according to plant water extracting abilities at different crop growing stages.'**

**was replaced by**

**'The use of these values was validated by Bonnemort et al. (1996), Voirin-Morel (2003) and Calvet et al. (2008). The idea behind this approach is that irrigation does not completely refill the soil, especially at the end of the growing season. Mechanical harvest requires relatively dry conditions to avoid soil compaction. The crop is allowed to use rainwater together with the initial available water content of the soil. This irrigation strategy allows the optimization of water withdrawal according to plant water extracting abilities at different crop growing stages.'**

**Comment R2.5:**

Lines 149-152: Please add that this first part describes sprinkler irrigation settings. Drip/flood description only starts in line 171.

**Response R2.5:**

**'The irrigation water flux is evenly distributed over a period of time of 8 hours ...'**

**was replaced by**

**'For sprinkler irrigation settings, the irrigation water flux is evenly distributed over a period of time of 8 hours ...'**

**Comment R2.6:**

Line 193: 20 non-irrigated + 20 irrigated * 3 types is 80

**Response R2.6:**

**Thanks for noting this. It was corrected.**

**Comment R2.7:**

Line 240: The random picking of harvest/emergence date seems to complicate things and you did not mention it previously. However it is relevant, because irrigation would not be allowed, if emergence is happening later or harvest earlier than the default date.

**Response R2.7:**

**We agree.**

**Comment R2.8:**

Lines 313-314: I suggest to change "simulated number of yearly irrigation events" to "simulated irrigation water amount". Events cannot be compared to amounts and you explain how the conversion is done in the next sentence.

**Response R2.8:**

**We agree. This sentence was rephrased accordingly.**

**Comment R2.9:**

Lines 360-364: Explain how you calculate the precipitation bias in Fig S4.6, either here in the text or in the figure caption. I assume it is ERA5 data minus weather station data for that pixel, right? You are arguing that the bias in ERA5 is the reason for too high simulated irrigation. But the absolute bias is also high in 2000 and 2005. If you want to take the relative change (2010 was a wetter year than 2005 and 2000) into account, I would think that this could be best seen in precipitation bias in [%] with respect to absolute precipitation. This should show higher deviations for 2010 than 2000 and 2005 and serve your point.

**Response R2.9:**

**Caption of Fig. S18 (ERA5 minus in situ observations) was changed accordingly. We tried to consider the percentage precipitation bias but this did not change the conclusions.**

**'In 2010, the ERA5 precipitation bias in July and August triggers a cumulated precipitation gap of 150 mm. The model responds to this water deficit by triggering irrigation, especially in August (Fig. S4.6c).'**

**was replaced by**

**'In 2010, the ERA5 precipitation bias from July to September triggers a cumulated precipitation gap of 103 mm (Fig. S18a). The model responds to this water deficit by triggering irrigation at the end of the growing season, especially in August (Fig. S4.6c). On the other hand, ERA5 is unbiased at the beginning of the growing period (May-June 2010).'**

**Comment R2.10:**

Lines 369-371: Where do I see this Boedhram data?

**Response R2.10:**

**'The data from Boedhram et al. (2001) show that…'**

**was replaced by**

**'Figure 2 in Boedhram et al. (2001) shows that…'.**

**Comment R2.11:**

Line 371: You could mention here that you will do a comparison across all "nature types" in the next section.

**Response R2.11:**

**'The satellite LAI observations are sensitive to both rainfed and irrigated vegetation.'**

**was replaced by**

**'The satellite LAI observations are sensitive to both rainfed and irrigated vegetation. A comparison across all vegetation types is presented in Section 3.3.'**

**Comment R2.12:**

Line 380: I assume you meant to say "without phenology (and without irrigation)" instead of "without irrigation" – the main difference here is the phenology, not the irrigation.

**Response R2.12:**

**Thanks for noting this. "without crop phenology" was added.**

**Comment R2.13:**

Lines 409-410: I would argue that the wider distributions are due to the effect of not having imposed emergence and harvest dates for natural vegetation.

**Response R2.13:**

**Yes.**

**'Compared to crop simulations, the experiments with crop phenology (ISBA_pheno and ISBA_pheno_irr) present earlier peak LAI dates, because rainfed vegetation affects the phenology.'**

**was replaced by**

**'Compared to irrigated crop simulations, the experiments with crop phenology (ISBA_pheno and ISBA_pheno_irr) present earlier peak LAI dates, because they include rainfed crops and natural vegetation. Emergence dates are not imposed to rainfed crops and to natural vegetation. This allows earlier leaf onset.'**

**Comment R2.14:**

Line 489: I would say, that rather than "empirical" it is "random".

**Response R2.14:**

**We agree. This sentence was rephrased accordingly.**

**Comment R2.15:**

Line 539: It is really hard to see the difference, please add a third panel with the difference to this figure.

**Response R2.15:**

**Yes. A new subfigure was added to Fig. S9 (Fig. S9c):**

[Figure]

**Comment R2.16:**

Table 1: List all possible values as well, not only the default. Irrigated "nature" type, I understand that the surface type is called this way in ISBA, but to call a crop "nature" sounds wrong to me. How about (at least in the paper), you rename it to "vegetation", or what it is: "land surface type". An irrigation water "turn" could be called "event". Explain the abbrv. "SWI" in the caption. I believe a "time lapse" is sth. else, how about "time interval"/"lapse of time"/"time span"?

**Response R2.16:**

**We revised Table 1 accordingly (see below).**

| Symbol | Definition | Range | Default value (this study) |
|---|---|---|---|
| $I_T$ | Irrigation type | Sprinkler, flood, and drip irrigation | sprinkler |
| $I_{NT}$ | Irrigated land surface type | All 20 land surface types (Fig. S1) | C3 crops, C4 crops, shrubs |
| $I_W$ | Water amount per irrigation event | 0 mm or more | 30 mm |
| $I_D$ | Irrigation event duration | 0.25 hour or more | 8 hours |
| $SWI_1$ | Soil wetness index threshold for triggering the first irrigation event | 0 to 1 | 0.70 |
| $SWI_2$ | Soil wetness index threshold for triggering the second irrigation event | 0 to 1 | 0.55 |
| $SWI_3$ | Soil wetness index threshold for triggering the third irrigation event | 0 to 1 | 0.40 |
| $SWI_{4+i}$ | Soil wetness index threshold for triggering the following irrigation events (i, integer > 0) | 0 to 1 | 0.25 |
| $\Delta t_{Wn}$ | Minimum time interval between two irrigation events (irrigation interval) | 0 days (e.g. drip irrigation) or more | 7 days |
| $\Delta t_{WH}$ | Minimum time interval between the last irrigation event and the harvest | 0 to 365 days | 15 days |
| $t_E$ | Emergence date | 1 January to 31 December | 15 May (± 15 days) |
| $t_H$ | Harvest date | 1 January to 31 December After emergence date | 15 September (± 15 days) |

**Comment R2.17:**

Figure 3: I suggest to use 5-year steps for the x-axis starting 1985.

**Response R2.17:**

**We revised this Figure accordingly:**

[Figure]

**Comment R2.18:**

Figure 7: This figure is never mentioned in the text.

**Response R2.18:**

**Figure 7 is mentioned in Section 3.3 ("A month by month analysis of the scores (Fig. 7) shows a significant improvement of *r* values in June and September").**